# EARTHSE: A BENCHMARK EVALUATING EARTH SCIENTIFIC EXPLORATION CAPABILITY FOR LARGE LANGUAGE MODELS

**Wanghan Xu**[1,2]∗ **Xiangyu Zhao**[2,3], **Yuhao Zhou**[2], **Xiaoyu Yue**[2], **Ben Fei**[2,4],
**Fenghua Ling**[2], **Wenlong Zhang**[2] †, **Lei Bai**[2] †
[1]Shanghai Jiao Tong University    [2]Shanghai Artificial Intelligence Laboratory
[3]The Hong Kong Polytechnic University    [4]The Chinese University of Hong Kong
[†]Corresponding author. {zhangwenlong,bailei}@pjlab.org.cn

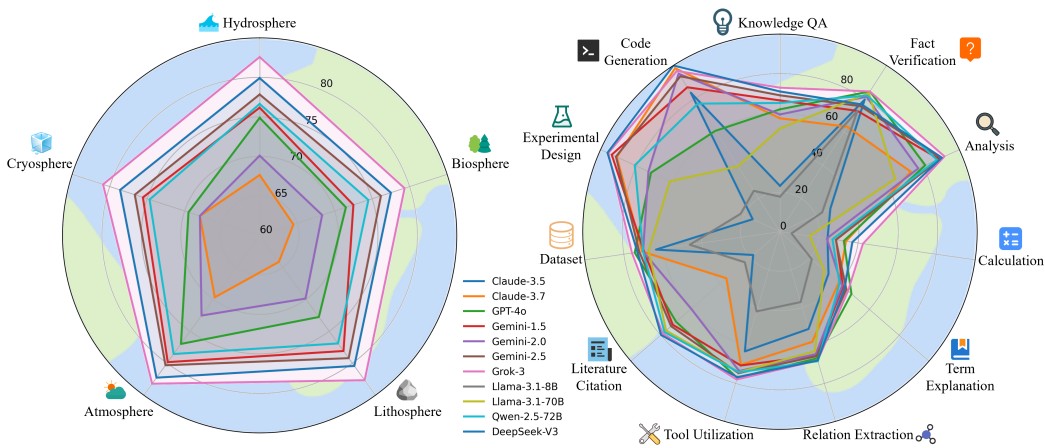

Figure 1: **Cross-domain Evaluation of Mainstream LLMs on EarthSE.** (a) EarthSE evaluates the capabilities of different LLMs in Earth's five spheres. (b) The multi-task evaluation demonstrates pronounced capability limitations in tasks such as calculation and term explanation.

## ABSTRACT

Advancements in Large Language Models (LLMs) drive interest in scientific applications, necessitating specialized benchmarks such as Earth science. Existing benchmarks either present a general science focus devoid of Earth science specificity or cover isolated subdomains, lacking holistic evaluation. Furthermore, current benchmarks typically neglect the assessment of LLMs' capabilities in open-ended scientific exploration. In this paper, we present a comprehensive and professional benchmark for the Earth sciences, designed to evaluate the capabilities of LLMs in scientific exploration within this domain, spanning from fundamental to advanced levels. Leveraging a corpus of 100,000 research papers, we first construct two Question Answering (QA) datasets: **Earth-Iron**, which offers extensive question coverage for broad assessment, and **Earth-Silver**, which features a higher level of difficulty to evaluate professional depth. These datasets encompass five Earth spheres, 114 disciplines, and 11 task categories, assessing foundational knowledge crucial for scientific exploration. Most notably, we introduce **Earth-Gold** with new metrics, a dataset comprising open-ended multi-turn dialogues specifically designed to evaluate the depth and diversity of LLMs in scientific exploration, including methodology induction, limitation analysis, and concept proposal. Extensive experiments reveal limitations in 11 leading LLMs across different domains and tasks, highlighting considerable room for improvement in their scientific exploration capabilities. The data is available on https://huggingface.co/ai-earth.

∗This work was done during his internship at Shanghai Artificial Intelligence Laboratory.

## 1 INTRODUCTION

In recent years, the rapid development of large language models (LLMs) has continuously pushed the boundaries of their capabilities, demonstrating remarkable performance in general knowledge question-answering tasks (Li et al., 2025). Against this backdrop, researchers are actively expanding the applications of LLMs to specialized and challenging scientific subfields (Zhang et al., 2023; Bi et al., 2023). This trend not only enhances the reasoning ability of LLMs for complex scientific problems but also holds potential for AI-assisted scientific discovery (Lu et al., 2024; Yamada et al., 2025). To systematically evaluate the performance of mainstream LLMs in scientific tasks, multiple science-oriented benchmarks (Wang et al., 2023; Rein et al., 2024) have been proposed.

However, a comprehensive benchmark for evaluating LLMs in the critical field of Earth science remains conspicuously absent. Existing benchmarks predominantly fall into two categories: a) general science benchmarks (e.g., ScienceQA (Lu et al., 2022), SciBench (Wang et al., 2023)) which lack the necessary specificity and depth in Earth science, often featuring questions of a common-sense nature. b) single-subdomain benchmarks (e.g., ClimaQA (Manivannan et al., 2024), OceanBench (Bi et al., 2023)) that concentrate on particular areas such as climate or ocean science, thus failing to encompass the broad and interdisciplinary spectrum of Earth science. Furthermore, the prevalent question-answering (QA) format in most benchmarks overlooks the evaluation of LLMs in open-ended scientific exploration tasks. Consequently, the construction of a *comprehensive* and *specialized* Earth science benchmark that incorporates assessments of *scientific exploration* capabilities represents an underexplored yet vital area of research.

In this paper, we present EarthSE, a comprehensive and specialized benchmark dataset for Earth science that uniquely incorporates evaluations of scientific exploration capabilities. To ensure both sufficient scale and high quality, we curate a corpus of over 100,000 Earth science academic papers as our primary data source. Through semantic analysis of titles and keywords, we categorize these papers into five major spheres (Manahan, 2006) and 114 sub-disciplines. Leveraging the publication venue and citation counts, we further stratify a subset of 10,000 papers into three distinct levels. The first level paper collection constitutes the largest portion, emphasizing comprehensiveness. The second level focuses on papers from high-impact journals, prioritizing specialized knowledge. The third level comprises highly cited papers, specifically designed for evaluating scientific exploration.

Scientific exploration demands multi-level competencies, spanning fundamental domain knowledge mastery to advanced critical reflection and innovative improvements. The former, amenable to explicit evaluation criteria, suits **question-answering** formats. The latter, lacking unified standards, is better assessed through **open-ended multi-turn dialogues**. Correspondingly, we develop two pipelines: one for QA data using predefined task formulations to directly generate high-quality pairs from papers, and another for dialogue data structuring papers to extract scientific inquiry workflows for multi-turn dialogues. Both pipelines include automated cleaning and human verification for quality.

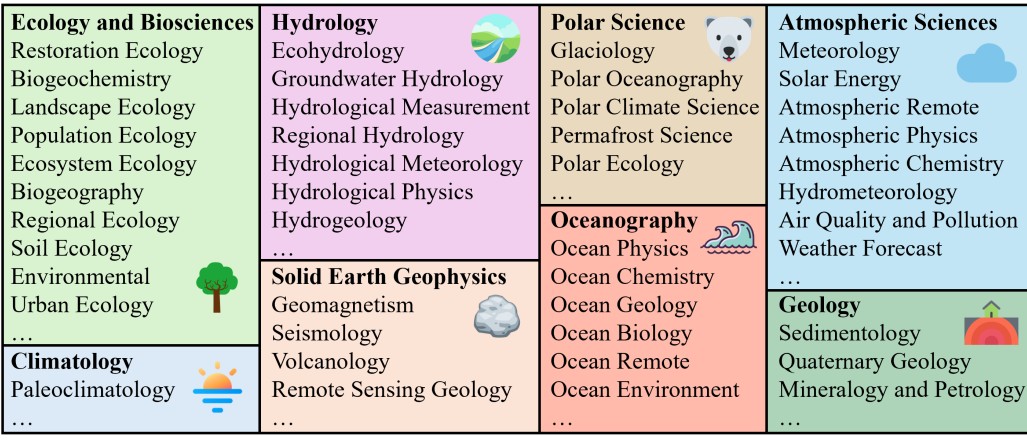

Figure 2: **EarthSE Covers 114 Disciplines in Earth Sciences.**

Consequently, we release three distinct datasets, each with unique characteristics. **Earth-Iron (QA)** encompasses 4133 questions across 11 tasks and 114 sub-disciplines (Figure 2), emphasizing foundational and broad scientific exploration capabilities. **Earth-Silver (QA)** prioritizes high-difficulty, specialized knowledge. **Earth-Gold (dialogue)** focuses on evaluating advanced scientific exploration capabilities such as summarization, reflection, and innovation within open-ended dialogues.

We summarize the contributions of this paper as follows:

- We built two QA datasets to evaluate the fundamental capabilities of scientific exploration: **Earth-Iron**, containing 4133 questions that span 114 subfields for broad assessment, and **Earth-Silver**, presenting more challenging and professional inquiries for deeper evaluation.
- We innovatively propose **Earth-Gold** and a new metric (i.e., SES) to assess advanced capabilities of scientific exploration (e.g., methodology induction, limitation analysis, and concept proposal) through **open-ended multi-turn dialogue**.
- Our systematic evaluation of 11 leading LLMs across these datasets reveals significant deficiencies in complex Earth science reasoning and open-ended scientific exploration.

## 2 RELATED WORK

**General Science Benchmark.** In recent years, LLMs have increasingly focused on scientific reasoning (Li et al., 2025), leading to the development of a range of benchmarks designed to assess model performance across diverse scientific domains (Thirunavukarasu et al., 2023; Zhang et al., 2024). General-purpose science benchmarks such as ScienceQA (Lu et al., 2022), SciBench (Wang et al., 2023) and MMLU-Pro (Wang et al., 2024b) cover disciplines from elementary to graduate levels. For instance, ScienceQA includes over 21,000 multi-modal questions spanning natural, social, and language sciences. MMLU-Pro extends MMLU (Hendrycks et al., 2020) with broader college-level subjects to evaluate deeper reasoning. SciBench emphasize college-level problem solving, drawing from canonical textbooks. Expert-curated datasets like GPQA (Rein et al., 2024), SuperGPQA (Du et al., 2025), and HLE (Phan et al., 2025) offer high-quality, graduate-level questions to assess fine-grained domain expertise. However, these benchmarks primarily prioritize breadth over depth, often underrepresenting Earth sciences. As a unique and complex discipline covering five spheres, Earth science requires more targeted evaluation.

**Specialized Subject Benchmark.** In addition to general benchmarks, several efforts have focused on subdomains within Earth sciences. For example, OceanGPT (Bi et al., 2023) introduces Ocean-Bench, a benchmark for oceanographic tasks such as QA, fact verification, and applied writing. ClimaQA (Manivannan et al., 2024) and ClimateWatch (Kraus et al., 2023) assess reasoning over climate data and models. ClimateBERT (Webersinke et al., 2021) and ClimaText (Varini et al., 2020) support stance detection and document classification in climate discourse. Geological (Deng et al., 2024) tasks are addressed by GeoBench, which supports entity recognition and relation extraction, and SeafloorAI (Nguyen et al., 2024), which evaluates multimodal reasoning over sedimentary and geomorphological features. While these benchmarks are domain-relevant, each focuses on a narrow component of the Earth, lacking a unified evaluation across the full Earth science spectrum.

Moreover, the majority of current benchmarks emphasize *question answering*, focusing on factual recall and reasoning over established knowledge. Yet, a critical frontier lies in enabling LLMs to support *scientific exploration and discovery*, a more open-ended capability involving summary and limitation analysis of existing methods, and hypothesis generation.

## 3 EARTHSE: EARTH SCIENCE EXPLORATION BENCHMARK

**Overview.** Scientific exploration requires capabilities from basic Earth science knowledge to advanced reflection and proposing new methods. Starting from 100,000 papers in the Earth sciences, we constructed three datasets. **Earth-Iron (QA)** contains 4133 questions covering 114 sub-disciplines, focusing on a comprehensive evaluation of fundamental scientific exploration abilities. **Earth-Silver (QA)** features more difficult and challenging questions, focusing on more specialized Earth science knowledge. **Earth-Gold (dialogue)** evaluates advanced scientific exploration abilities through open-ended dialogue, focusing on reflecting on existing problems and proposing new approaches.

Table 1: **Comparison between Existing Benchmarks and EarthSE.** Comparison across (1) question volume, (2) data sources, (3) Earth sphere coverage, (4) graduate-level difficulty, (5) multiple subsets and (6) scientific exploration assessment. EarthSE uniquely offers comprehensive domain coverage, professional difficulty, and evaluation of scientific exploration in Earth sciences.

| Benchmark | #Ques. | Source | Earth Cover | Grad-Diff | Multi-Sub | Scientific Exploration |
|---|---|---|---|---|---|---|
| ScienceQA (Lu et al., 2022) | 21,208 | High School Courses | ✗ | ✗ | ✗ | ✗ |
| MMLU-Pro (Wang et al., 2024b) | 12,032 | Exam Questions | ✗ | ✗ | ✗ | ✗ |
| SciBench (Wang et al., 2023) | 869 | 10 Textbooks | ✗ | ✗ | ✗ | ✗ |
| GPQA (Rein et al., 2024) | 448 | Expert Curated | ✗ | ✓ | ✓ | ✗ |
| OceanBench (Bi et al., 2023) | 12,426 | Domain Texts | ✗ | ✓ | ✗ | ✗ |
| ClimaQA (Manivannan et al., 2024) | 3502 | 18 Textbooks | ✗ | ✓ | ✓ | ✗ |
| GeoBench (Deng et al., 2024) | 2439 | Exam Questions | ✗ | ✓ | ✗ | ✗ |
| **EarthSE** | **4133** | **100,000 Earth Science Papers** | ✓ | ✓ | ✓ | ✓ |

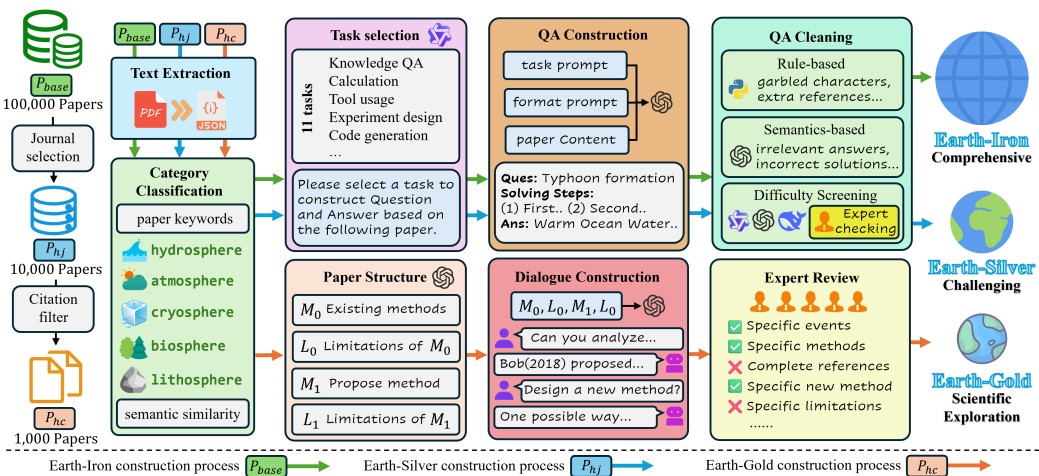

Figure 3: **Construction Process of EarthSE.** Automated construction of three-tiered benchmarks (Earth-Iron, Earth-Silver, Earth-Gold) from 100K papers, enabling comprehensive evaluation of knowledge coverage, professional proficiency, and scientific exploration capabilities.

## 3.1 TASK DEFINITION FROM FOUNDATIONAL TO ADVANCED

**Foundational Science Task Definition.** To comprehensively evaluate the capabilities of LLMs in Earth exploration tasks, spanning from foundational to advanced levels, we defined 11 fundamental research tasks, as detailed in Table 2. These tasks cover a broad spectrum, ranging from basic understanding and reasoning to advanced research skills. These capabilities form the foundation for LLMs to conduct scientific exploration. For example, the "Dataset" task specifically focuses on datasets within the Earth sciences. Given the significant diversity of data (such as one-dimensional seismic wave data and two-dimensional remote sensing images) across the numerous sub-disciplines of Earth science, enhancing LLMs' understanding of these varied datasets is crucial.

**Scientific Exploration Task Definition.** Beyond these 11 fundamental scientific exploration tasks, we have also defined a scientific discovery task formulated as open-ended dialogue. Through systematic analysis of these papers, we identify a recurrent research pattern: "analyzing limitations of existing work → proposing novel methods" (Yang et al., 2024b). This inspires our formalization of scientific exploration as an iterative self-negation process, mathematically expressed as: $(M^{i+1}, L^{i+1}) = \text{LLM}(M^i, L^i)$, where $M$ denotes methodology, $L$ represents limitation analysis of $M$, and $i$ indicates dialogue turns. This recursive framework simulates the human scientific process of critically examining prior work's constraints and progressively improving upon them. Consequently, models with genuine scientific discovery potential must demonstrate robust self-critique and self-improvement capabilities (Li, 2024).

Table 2: **Foundational Task Categories and Descriptions.**

| | Task | Description |
|---|---|---|
| **Understanding** | Term Explanation (Heiman, 2025) | Requires defining technical concepts (e.g., "Define 'medium-range weather forecast' and explain its significance"). |
| | Knowledge QA (Xu et al., 2024) | Requires factual explanations with detailed descriptions of distributions or characteristics (e.g., "Describe the distribution of fishery resources in the East China Sea"). |
| | Fact Verification (Zhang & Gao, 2023) | Involves validating claims' accuracy (e.g., "Verify whether the reported sea level rise data is correct"). |
| **Reasoning** | Analysis (Cai et al., 2024) | Demands logical reasoning to draw conclusions from data (e.g., "Determine which evidence supports a given conclusion"). |
| | Relation Extraction (Li et al., 2024) | Needs analysis of interconnections between entities (e.g., "Summarize ecological relationships between butterflies and plants"). |
| | Calculation (Stephan et al., 2024) | Involves multi-step mathematical operations with numerical answers (e.g., "Calculate mean annual precipitation from the dataset"). |
| **Research** | Tool Utilization (Yuan et al., 2024) | Involves recommending domain-specific methodologies (e.g., "Suggest atmospheric models for weather prediction"). |
| | Literature Citation (Byun et al., 2024) | Demands proper academic references (e.g., "List key publications on ocean circulation including citations like (Ravuri et al., 2021)"). |
| | Dataset (Taylor et al., 2022) | Requires introducing or recommending research datasets (e.g., "Recommend ERA5 (Hersbach et al., 2020) for medium-range weather forecasting"). |
| | Experimental Design (Chen et al., 2024) | Needs detailed methodological planning (e.g., "Design an experiment to investigate soil moisture's impact on photosynthesis"). |
| | Code Generation (Gu, 2023) | Involves writing functional code (e.g., "Visualize rainfall data using Python's Matplotlib library (Bisong, 2019)"). |

## 3.2 PAPER CORPUS COLLECTION

The study uses Earth science academic papers as data because: (a) their dense, high professional knowledge facilitates quality data creation; (b) their structured format aligns with general scientific discovery processes. During collection, we obtain and convert 100,000 PDFs to structured JSON using MinerU (Wang et al., 2024a). Semantic similarity (Devika et al., 2021) on abstracts and Earth sphere keywords (see Figure 7) classifies papers into five Earth spheres, as detailed in Table 3.

We define the initial collection of 100,000 papers as the base dataset $P_{base}$. From this collection, we first select a subset $P_{hj}$ comprising 10,000 papers published in high-impact Earth science journals (see Table 7 for details). We then extract the top 10% most cited papers from $P_{hj}$ to form the high-citation core dataset $P_{hc}$ with 1,000 papers. The entire selection process maintains strict balance across all five spheres, with detailed distributions provided in Table 3. Figure 4 further illustrates the citation distribution patterns across different spheres in the high-citation dataset $P_{hc}$. $P_{base}$, $P_{hj}$ are used to construct Earth-Iron, Earth-Silver, Earth-Gold respectively, as depicted in Figure 3.

Table 3: **Number of Papers on the Five Spheres of Earth.**

| Earth Sub-domain | $P_{base}$ | $P_{hj}$ | $P_{hc}$ |
|---|---|---|---|
| Biosphere | 21,248 | 1,554 | 201 |
| Lithosphere | 22,820 | 2,357 | 236 |
| Atmosphere | 24,213 | 2,401 | 240 |
| Hydrosphere | 23,425 | 2,254 | 226 |
| Cryosphere | 11,402 | 1,969 | 217 |
| Total | 103,108 | 10,535 | 1,120 |

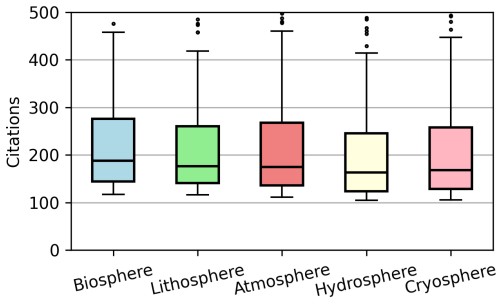

Figure 4: **Citations of Papers in $P_{hc}$.**

## 3.3 EARTH-IRON/SILVER: QA BENCHMARK FOR FOUNDATIONAL TASKS

Earth-Iron and Earth-Silver are constructed from $P_{base}$ and $P_{hj}$ respectively, using the same pipeline as shown in the first row of Figure 3. Characteristically, Earth-Iron offers a large and comprehensive set of questions, while Earth-Silver features more difficult and specialized questions.

**Task Selection.**    Before QA construction, a small LLM analyzes each paper's abstract to determine the most suitable task type from 11 foundational tasks shown in Table 2 for QA generation. This content-aware assignment enhances question relevance. For instance, papers containing substantial numerical results preferentially generate computation questions.

**QA Construction.**    In the question generation phase, paper content and task prompts are fed into GPT-4 (Achiam et al., 2023) to produce QA pairs. To ensure answer accuracy, the model is required to provide complete Chain-of-Thought (CoT) (Wei et al., 2022) reasoning that includes both the final answer and its derivation process. These CoTs not only form integral parts of the questions but also serve as critical references for subsequent data cleaning, effectively mitigating errors caused by LLM hallucinations (Wei et al., 2024).

**Data Cleaning.**    The data cleaning process employs a dual-phase strategy: (a) rule-based primary cleaning for formatting issues (missing options, irregular answers, improper references); (b) semantics-based advanced cleaning using LLMs to detect deeper problems (multiple correct options, irrelevant/incorrect answers). Throughout this process, CoT reasoning plays a pivotal role, as its explicit step-by-step derivation provides reliable evidence for error detection.

After data cleaning, we performed difficulty screening using mainstream LLMs. Based on testing results from both proprietary and open-source models, we removed questions with accuracy rates exceeding 80% to ensure sufficient overall challenge. For questions with accuracy rates between 60% and 80%, human experts determined their retention based on the question's value.

### 3.4    Earth-Gold: Open-end Dialogue Benchmark for Exploration Task

Most existing benchmarks predominantly employ question-answering formats, which, while effective for knowledge assessment, fail to capture the open-ended exploration paradigm characteristic of genuine scientific research. To evaluate LLMs' capabilities in open scientific exploration, we construct the Earth-Gold dataset derived from $P_{hc}$, a collection of 1,000 highly-cited Earth science papers, as shown in the second row of Figure 3. These papers exemplify superior scientific reasoning patterns, making them ideal prototypes for studying scientific exploration behaviors.

**Paper Structurization**    Following the scientific exploration task definition in Section 3.1, we decompose each paper in $P_{hc}$ into four structured components:

- **Existing Method Summary** ($M^0$): Extracted from related work sections, providing comprehensive synthesis of current methodologies.
- **Limitation Analysis** ($L^0$): Derived from motivation sections, identifying precise shortcomings that constitute the starting point for new research.
- **Novel Method Proposal** ($M^1$): Abstracted from methods sections, capturing the core innovations.
- **New Method Limitations** ($L^1$): Distilled from discussion sections, anticipating potential constraints of proposed solutions.

**Dialogue Construction**    With $M^0$, $L^0$, $M^1$, and $L^1$ extracted, we employ GPT-4 to generate two-turn dialogues simulating human-AI collaborative scientific exploration. The first turn requires the AI assistant to summarize existing methods and critically analyze their limitations given a research direction; the second turn directs the assistant to propose improved methods addressing these limitations while objectively assessing the new methods' potential constraints. This dialogue structure authentically replicates human ideation processes assisted by AI, demanding both extensive scientific knowledge and critical thinking abilities from LLMs.

**Human Expert Validation**    Following dialogue generation, domain experts conduct rigorous quality control using multi-criteria evaluation: a) **Information Density Scoring**: Higher scores are assigned to dialogues containing specific analytical methods, datasets, or well-defined events (e.g., earthquakes, typhoons), while vague discussions receive lower scores. b) **Methodological Quality Assessment**: Proposed methods receive higher scores if they are systematically enumerated and

concretely implementable. Non-specific proposals are filtered out. Through this stringent validation process, we curate the final Earth-Gold dataset, which exclusively comprises high-quality scientific exploration dialogues, thus establishing a new benchmark for evaluating LLMs' research capabilities.

**A New Metric: Scientific Exploration Score (SES).** Earth-Gold evaluates open-ended scientific dialogue, where standard correctness metrics are inadequate due to the inherent diversity of valid scientific exploration beyond definitive answers. To address this, we innovatively measure performance using retention rate and diversity. For each user question in the dialogues, the LLM generates $M$ diverse responses (temperature=0.6) (Peeperkorn et al., 2024), and then we compute:

- **Retention Rate ($r$)**: GPT-4 ranks $M$ generated answers and the reference answer by reflective depth and innovation. Retention rate $r = (i-1)/M$ (where $i$ is the reference answer rank) quantifies the model's preservation of high-quality scientific reasoning. If $r = 0$, it means that all LLM answers are worse than the reference answers.

- **Diversity ($d$)**: We compute embeddings $\mathbf{v}_i$ for each response using sentence-transformers, then derive the mean vector $\bar{\mathbf{v}}$. The average cosine similarity $\bar{s}$ between $\bar{\mathbf{v}}$ and $\mathbf{v}_i$ is calculated as Equation 1. Since lower similarity indicates higher diversity, define $d = 1/\bar{s}$.

$$\bar{s} = \frac{1}{M}\sum_{i=1}^{M}\cos(\mathbf{v}_i, \bar{\mathbf{v}}), \text{ where } \bar{\mathbf{v}} = \frac{1}{M}\sum_{i=1}^{M}\mathbf{v}_i. \tag{1}$$

The Scientific Exploration Score (SES) combines retention ($r$) and diversity ($d$) multiplicatively: SES $= r \times d$. Since $\bar{s}$ is relatively close to 1, we normalize $\bar{s}$ to $[0.9, 1]$ in actual calculations for better comparison, that is, SES $= \frac{r}{10 \times (\bar{s} - 0.9)}$. Higher SES values indicate superior open-ended scientific thinking, reflecting both answer quality (retention) and conceptual breadth (diversity).

## 4 EXPERIMENT

### 4.1 EXPERIMENTAL SETUP

This paper introduces three evaluation datasets: Earth-Iron and Earth-Silver as question answering (QA) datasets, and Earth-Gold as an open-ended dialogue dataset. The QA datasets incorporate 4 question formats: multiple-choice (MC), fill-in-the-blank (FIB), true/false (TF), and free-response (FR) questions. For MC, FIB, and TF questions, we use accuracy (ACC) as the metric; for FR questions, we employ the win rate (WR) against the reference answer evaluated by GPT-4, and semantic similarity (SS) as metrics.

**Accuracy (Acc.)** For closed-form questions (multiple-choice, fill-in-the-blank, and true/false) with deterministic correct answers, we calculate accuracy as the exact match rate between model outputs and reference solutions.

**Win Rate (WR)** (Bi et al., 2023) For free-response questions, we employ GPT-4 as an impartial judge to compare model answers against reference answers along three dimensions: relevance (degree of topical alignment, penalizing tangential responses), scientific rigor (logical consistency and factual correctness), and specificity (depth of targeted analysis, penalizing generic statements). The evaluation protocol ensures fairness by blinding GPT-4 to answer origins. Win rate quantifies the percentage of cases where the evaluated model's response surpasses the reference.

**Semantic Similarity (SS)** (Devika et al., 2021) For free-response questions, we additionally measure the conceptual alignment between model outputs and reference answers using sentence embeddings. Specifically, we utilize sentence-transformers to project responses into a 384-dimensional semantic space, then compute their cosine similarity as the semantic similarity metric.

For Earth-Gold, we use the SES defined in Section 3.4 to evaluate the performance of scientific exploration dialogues.

Table 4: **Evaluation Metrics for Leading Models on the Earth-Iron and Earth-Silver Benchmarks.** For MC (Multiple Choice), TF (True/False), and FIB (Fill-in-the-Blank) questions, Accuracy (ACC) is used as the evaluation metric, reported in percentage (%). For FR (Free Response) questions, Win Rate (WR), Semantic Similarity (SS) are employed as an evaluation metrics.

| Model | Earth-Iron (More Comprehensive) | | | | | Earth-Silver (More Challenging) | | | | |
| --- | --- | --- | --- | --- | --- | --- | --- | --- | --- | --- |
| | MC↑ | TF↑ | FIB↑ | FR-WR↑ | FR-SS↑ | MC↑ | TF↑ | FIB↑ | FR-WR↑ | FR-SS↑ |
| Llama-3.1-8B (Grattafiori et al., 2024) | 59.41 | 74.36 | 2.52 | 13.70 | 0.76 | 36.00 | 54.00 | 2.02 | 4.40 | 0.72 |
| Llama-3.1-70B (Grattafiori et al., 2024) | 91.56 | 87.91 | 6.63 | 61.85 | 0.80 | 56.00 | 63.60 | 4.00 | 18.40 | 0.80 |
| Qwen-2.5-72B (Yang et al., 2024a) | 92.42 | 86.26 | 11.96 | 92.05 | 0.79 | 53.60 | 64.40 | 9.20 | 44.40 | 0.78 |
| DeepSeek-V3 (Liu et al., 2024) | 93.40 | 81.14 | 18.99 | 97.60 | **0.81** | 58.00 | 56.40 | 12.80 | 75.20 | **0.81** |
| GPT-4o (OpenAI, 2024) | 93.28 | 88.28 | 19.12 | 82.00 | **0.81** | 55.60 | 69.60 | **18.40** | 22.00 | 0.80 |
| Gemini-1.5-Flash (Team et al., 2024) | 90.83 | 75.82 | 13.65 | 95.60 | 0.79 | 54.40 | 44.80 | 8.00 | 62.80 | 0.78 |
| Gemini-2.0-Flash (Team et al., 2023) | 92.67 | 87.55 | 14.69 | 77.10 | 0.77 | 54.40 | **72.40** | 11.60 | 34.80 | 0.75 |
| Gemini-2.5-Flash (G.Comanici et al., 2025) | 93.15 | 77.84 | 17.02 | 95.81 | 0.75 | 58.00 | 55.60 | 13.65 | 74.30 | 0.75 |
| Claude-3-5-Haiku (Kurokawa et al., 2024) | 91.08 | 83.52 | 12.48 | 12.05 | 0.79 | 56.80 | 60.80 | 9.60 | 4.40 | 0.77 |
| Claude-3-7-Sonnet (Lim et al., 2025) | **94.01** | 61.90 | 20.68 | 75.00 | 0.80 | **62.40** | 41.20 | 17.20 | 28.40 | 0.79 |
| Grok-3 (de Carvalho Souza & Weigang, 2025) | 93.03 | **88.64** | **21.85** | **98.70** | **0.81** | 53.20 | 70.40 | 15.20 | **83.60** | **0.81** |
| Mean | 89.53 | 81.20 | 14.50 | 72.86 | 0.78 | 54.40 | 59.38 | 11.06 | 41.15 | 0.77 |

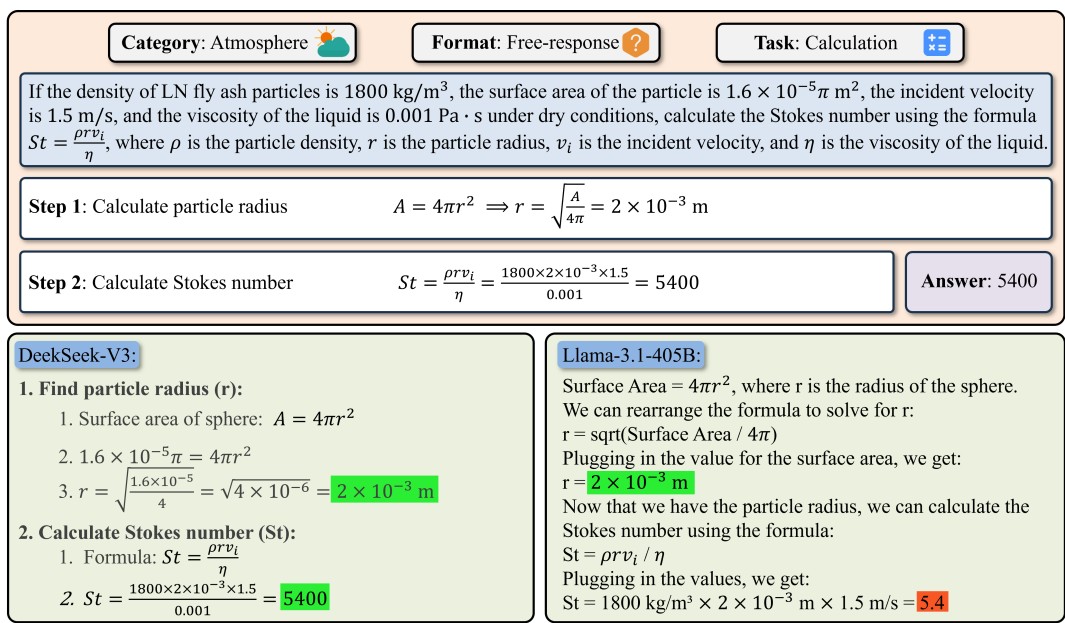

Figure 5: **Case Study from Earth-Silver.** Some models make mistakes when applying formulas in multiple steps, resulting in low accuracy in calculation questions.

## 4.2 EARTH-IRON/SILVER: ASSESSING BROAD FOUNDATIONAL CAPABILITIES

Earth-Iron is a comprehensive QA benchmark consisting of 4133 questions spanning the Earth sciences domain. Table 4 presents comparative performance metrics across these question formats for various LLMs. Most models perform well on multiple-choice questions but struggle with fill-in-the-blank tasks. In free-response questions, performance varies significantly; some models (e.g., Claude-3.5) provide overly general answers, resulting in poor Win Rate against the reference answers.

We compute each model's overall competency across Earth's five spheres, as detailed in Figure 1. This figure visually illustrates the capability distribution of different models across the five spheres. Notably, Grok-3 achieves state-of-the-art (SOTA) performance across all spheres. While most models exhibit a relatively balanced capability distribution, such as DeepSeek-V3, Gemini-2.5, some show specific weaknesses, for example, GPT-4o in the cryosphere.

The right panel in Figure 1 illustrates the capabilities of different models across the 11 fundamental scientific tasks. Most models exhibit a similar capability distribution, particularly the SOTA LLMs, likely due to similar training paradigms. Notably, most models perform relatively poorly on the calculation task. Additionally, the term explanation metric is generally low, indicating a potential weakness in LLMs' understanding of specialized Earth science terminology.

Compared to Earth-Iron, Earth-Silver is constructed from a higher-quality corpus of original papers, resulting in a significantly more challenging dataset. The experimental results on the right side of Table 4 show that most LLMs achieve only 54% accuracy on multiple-choice questions (compared to a 25% random chance rate). This not only reflects the high difficulty of Earth-Silver but also indicates considerable room for improvement in the performance of current LLMs within the Earth sciences. On fill-in-the-blank tasks, the average performance of LLMs is notably low, at just 11%.

Figure 5 presents representative examples from these challenging tasks alongside model responses. The case study analysis reveals frequent formula misuse and computational errors in Earth science-specific calculations, indicating critical knowledge gaps in domain-specific reasoning.

## 4.3 EARTH-GOLD: EVALUATING OPEN-END SCIENCE EXPLORATION CAPABILITY

Earth-Gold, a core innovation, evaluates LLMs in open scientific exploration dialogues using our novel Scientific Exploration Score (SES), which assesses divergent thinking (diversity $d$) and answer quality (retention rate $r$). Table 5 shows performance with each model generating $M = 3$ responses for $r$ and $d$ calculation. The results indicate that most models achieve a retention rate of less than 50%, suggesting that over half of the generated responses underperform the reference answers. Regarding the diversity metric, most models exhibit low diversity scores. This indicates that when responding to open-ended questions, LLMs tend to generate similar answers across multiple attempts, which significantly contrasts with the divergent thinking characteristic of human scientists during scientific exploration.

Table 5: **Scientific Exploration Capabilities of Mainstream Models on Earth-Gold.** Earth-Gold assesses LLMs' research potential through open-ended scientific dialogues, employing three metrics: Retention (proportion of responses outperforming references), Diversity (measure of divergent thinking), and their composite Scientific Exploration Score (SES).

| Model | Retention (%) ↑ | Diversity ↑ | SES ↑ |
|---|---|---|---|
| Llama-3.1-8B | 8.00 | **3.9813** | 0.3301 |
| Llama-3.1-70B | 11.78 | 1.4891 | 0.2453 |
| Qwen-2.5-72B | 7.11 | 1.7158 | 0.1375 |
| DeepSeek-V3 | 38.00 | 1.6942 | 0.6599 |
| GPT-4o | 9.44 | 1.0347 | 0.0981 |
| Gemini-1.5 | 19.67 | 1.4437 | 0.1989 |
| Gemini-2.0 | 18.22 | 2.6290 | 0.6505 |
| Gemini-2.5 | **50.56** | 2.7016 | **1.3710** |
| Claude-3.5 | 14.67 | 1.5517 | 0.2396 |
| Claude-3.7 | 31.89 | 1.7130 | 0.5465 |
| Grok-3 | 17.22 | 1.5284 | 0.2727 |

Our analysis identifies three primary issues in these subpar responses: (1) overly generic content lacking specific details, (2) non-specific analyses of limitations, and (3) excessively broad proposed solutions. Figure 6 provides a visual comparison between high-quality and low-quality responses.

## 4.4 CoT GUIDANCE ENHANCES PERFORMANCE ON CHALLENGING QUESTIONS

FIB ACC in Table 4 show lower accuracy, likely due to their inherent difficulty compared to MC/TF. Since our question construction generates step-by-step explanations resembling CoT reasoning, we investigate if providing LLMs with initial CoT steps during inference improves performance on these challenging questions.

Table 6 illustrates the model's accuracy with varying numbers of provided CoT steps. As is evident, increasing the number of CoT steps offered as hints significantly improves the model's accuracy. This suggests that for more challenging questions, even without modifying the model's fundamental capabilities, leveraging CoT-augmented reasoning at inference time can substantially boost performance. This finding provides a promising foundation for inference-time scaling strategies.

Table 6: **The Impact of Using CoT Guidance on FIB ACC.**

| Inspired CoT Steps | FIB (%) ↑ |
|---|---|
| DeekSeek-V3 | |
| 0 (Baseline) | 12.80 |
| 1 | 21.60 |
| 2 | 29.60 |
| 3 | **45.60** |
| GPT-4o | |
| 0 (Baseline) | 18.40 |
| 1 | 25.60 |
| 2 | 38.80 |
| 3 | **55.60** |

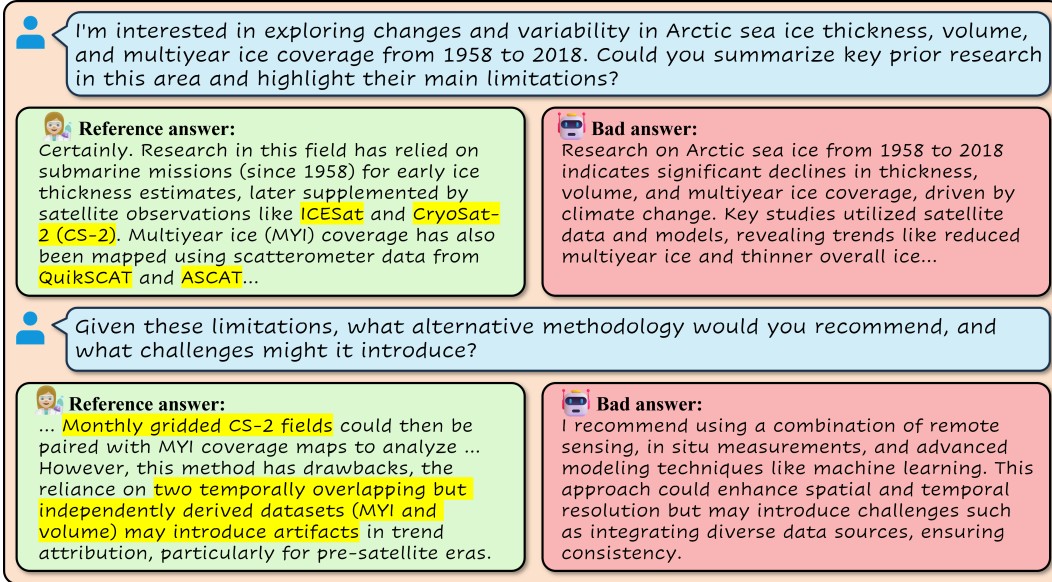

Figure 6: **Case Study from Earth-Gold.** Earth-Gold is used to evaluate the performance of LLMs in open scientific exploration. For the same user questions, the reference answers contain more details, such as the yellow text, while the bad answers are very general.

## 5 CONCLUSION

The increasing interest in leveraging Large Language Models (LLMs) for scientific applications underscores the need for specialized benchmarks, particularly in fundamental yet multifaceted domains like Earth science, which current general or fragmented benchmarks inadequately address, especially regarding open-ended scientific exploration. To this end, this paper introduces a comprehensive and professional Earth science benchmark comprising two QA datasets, Earth-Iron and Earth-Silver, and a novel open-ended dialogue dataset, Earth-Gold, built upon a 100,000-paper corpus. Earth-Iron provides broad foundational assessment across five Earth spheres, 114 disciplines, and 11 tasks, while Earth-Silver offers challenging, professional-level questions. Notably, Earth-Gold evaluates advanced scientific exploration abilities through multi-turn dialogues. Experiments show LLMs perform reasonably on basic Earth-Iron QA but significantly worse on challenging Earth-Silver. Earth-Gold reveals below low retention and diversity in open-ended dialogues. These quantitative results highlight current LLMs' limitations in Earth science knowledge depth and genuine scientific exploration, indicating significant room for improvement.

The limitation of this work lies in the fact that it does not integrate the 11 tasks to create a more complex task chain, highlighting a gap in achieving automated scientific discovery.

## ETHICS AND REPRODUCIBILITY STATEMENT

The data used in this study were sourced from OpendataLab (He et al., 2024), and were collected in strict accordance with relevant guidelines, avoiding any sources that restrict copying or redistribution. This work aims to establish a benchmark for evaluating the scientific discovery capabilities of large language models in Earth sciences. We believe that this research not only promotes the responsible use of artificial intelligence in scientific investigations but also enhances public understanding of Earth sciences. All data are released anonymously, with the full dataset publicly available on Anonymous HuggingFace (`https://huggingface.co/ai-earth`). The details of data processing are described in detail in the Appendix.

## ACKNOWLEDGMENTS

This work is supported by Shanghai Artificial Intelligence Laboratory.

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

# A   APPENDIX

## A.1   PAPER SELECTION AND TIERED CLASSIFICATION

The proposed benchmark derives from 100,000 papers downloaded from OpenDataLab (He et al., 2024), which undergo a tiered classification process resulting in three distinct paper collections:

- Base paper collection ($P_{\text{base}}$)
- High-journal-impact papers ($P_{\text{hj}}$)
- High-citation papers ($P_{\text{hc}}$)

The $P_{\text{base}}$ collection is constructed through semantic similarity analysis of paper titles and keywords. Our methodology employs both positive and negative keywords to enhance selection precision. Negative keywords effectively filter out semantically related but thematically irrelevant papers. For instance, when using "earth" as a positive keyword, astronomy-related papers may incorrectly appear relevant. To address this, we introduce "cosmos" as a negative keyword, requiring papers to maintain a similarity score below a specified threshold (0.1) with negative keywords while exceeding a minimum threshold (0.2) with positive keywords. This dual-threshold approach yields the final $P_{\text{base}}$ collection of 100,000 papers.

Using an analogous positive and negative keyword approach, we accurately classify papers into five Earth science subdomains (hydrosphere, biosphere, lithosphere, atmosphere, and cryosphere). The complete domain-specific keyword system is presented in Figure 7.

---

**Positive and Negative Keywords**

**Overall Positive Keywords**
Earth, Earth system, hydrosphere, biosphere, lithosphere, atmosphere, cryosphere.
**Overall Negative Keywords**
cell biology, virus, pharmaceuticals, chemistry, physics, astronomy, food science, proteins, microbiology.

- - - - - - - - - - - - - - - - - - - - - - - - - - - - - - - - - - - - - - - - - - - - - - - -

**Hydrosphere Positive Keywords**
water cycle, ocean, rivers, lakes, groundwater, ice caps, aquifers, precipitation, evaporation, humidity.
**Hydrosphere Negative Keywords**
chemistry, universe, planets, astronomy, astrophysics, space, stars, galaxy, cosmology.

- - - - - - - - - - - - - - - - - - - - - - - - - - - - - - - - - - - - - - - - - - - - - - - -

**Biosphere Positive Keywords**
ecosystem, biodiversity, habitat, species, biomes, ecological balance, carbon cycle.
**Biosphere Negative Keywords**
cell biology, chemistry, medicine, pharmacology, microbiology, biochemistry, toxicology, pathology, clinical.

- - - - - - - - - - - - - - - - - - - - - - - - - - - - - - - - - - - - - - - - - - - - - - - -

**Lithosphere Positive Keywords**
earthquake, tectonic plates, earth's crust, minerals, rocks, soil, sediments, mountains, volcanoes, landforms, geological processes.
**Lithosphere Negative Keywords**
ancient texts, archaeology, culture, history, artifacts, civilization, prehistoric, mythology, anthropology.

- - - - - - - - - - - - - - - - - - - - - - - - - - - - - - - - - - - - - - - - - - - - - - - -

**Atmosphere Positive Keywords**
stratosphere, troposphere, weather, climate, greenhouse gases, ozone layer, air pressure, humidity, winds, carbon dioxide, temperature.
**Atmosphere Negative Keywords**
universe, galaxy, astronomy, astrophysics, space, stars, planets, cosmology, black holes, nebula, solar system.

- - - - - - - - - - - - - - - - - - - - - - - - - - - - - - - - - - - - - - - - - - - - - - - -

---

**Cryosphere Positive Keywords**
glaciers, ice sheets, sea ice, permafrost, snowpack, icebergs, frozen ground, climate change, albedo effect, polar regions.
**Cryosphere Negative Keywords**
frozen food, ice cream, refrigeration, freezing, cold storage, ice cubes, food preservation, chilling, frost.

Figure 7: **Positive and Negative Keywords.**

From the initial paper collection $P_{\text{base}}$, we select a subset of papers $P_{\text{hj}}$ published in high-quality journals based on multiple criteria including impact factor (IF) (Garfield et al., 1994), disciplinary ranking, and expert assessment. Representative journals are listed in Table 7, with the complete journal inventory provided in Supplementary Materials.

Table 7: **Partial List of Selected Journal Titles.**

| CELL | NATURE | SCIENCE | NATURE BIOTECHNOLOGY |
|---|---|---|---|
| NATURE CELL BIOLOGY | NATURE CHEMICAL BIOLOGY | NATURE CHEMISTRY | NATURE CLIMATE CHANGE |
| NATURE COMMUNICATIONS | NATURE DIGEST | NATURE GENETICS | NATURE GEOSCIENCE |
| NATURE IMMUNOLOGY | NATURE MATERIALS | NATURE MEDICINE | NATURE METHODS |
| NATURE NANOTECHNOLOGY | NATURE NEUROSCIENCE | NATURE PHOTONICS | NATURE PHYSICS |
| NATURE REVIEWS CANCER | NATURE REVIEWS CARDIOLOGY | NATURE REVIEWS CLINICAL ONCOLOGY | ... |

Finally, we select the 1000 papers $P_{\text{hc}}$ with the highest number of citations from $P_{\text{hj}}$ to form a collection. The distribution of papers in each collection is shown in Table 3.

## A.2   QA CONSTRUCTION

When we construct QA questions, the prompts we use will vary depending on the format and task of the question. For questions of different formats, we use the following prompts.

Prompt for Different Question Formats

**Free Form QA**
Free-form questions and answers allow for open-ended responses. These questions typically require detailed explanations, recommendations, or descriptions. The answers can vary in length and structure, depending on the complexity of the question. For example, a question about improving the marine environment may require a list of measures with explanations. Format Rules: Answers can be in any format, including sentences, lists, or paragraphs. There are no strict restrictions on length or structure.

----------------------------------------------------------------

**Multiple Choice**
Multiple-choice questions provide a question followed by several answer choices, typically labeled with letters (e.g., A, B, C, D). Only one choice is correct. The task is to select the correct choice by providing the corresponding letter. Format Rules: The question requires a prompt along with four options, one of which is the correct answer. The answer must be a single letter corresponding to the correct choice (e.g., 'A', 'B', 'C', or 'D'). The answer should not include the full text of the choice.

----------------------------------------------------------------

**True False**
True/false questions require determining whether a given statement is correct or incorrect. The answer must be either 'True' or 'False', depending on the accuracy of the statement. Format Rules: The answer must be exactly 'True' or 'False'. No additional text or explanations are allowed.

----------------------------------------------------------------

---

**Free Form QA**

Fill-in-the-blank questions provide a sentence or statement with one or more missing words, phrases or number. The task is to complete the sentence by filling in the blank(s) with the correct word(s). Format Rules: The answer must be the exact word or phrase that fits the blank. No additional text or explanations are allowed unless explicitly requested.

---

Figure 8: **Prompt for Different Question Formats.**

The prompts for different tasks are shown in Table 2. The complete QA prompt is shown in Figure 9.

---

**Complete QA Construction Prompt**

**System Prompt**
You are a highly skilled scientist with a deep expertise in reading and analyzing scientific literature. Below is a research paper that you will carefully examine. Your task is to generate a well-structured question and answer based on the provided instructions. Ensure that your output is precise, relevant, and adheres to the specified guidelines.
**Instructions**
1. **Format**: <Format Prompt>
2. **Task**: <Task Prompt>
3. **Answer Explanation**: Each question must be accompanied by a corresponding answer explanation.
4. **Output Format**: The output should be structured as a dictionary, including the following keys:
- **question**: The generated question.
- **answer**: The correct answer to the question.
- **explanation**: A list containing the explanation(s) for the answer.
5. **Select One Specific Detail**
Do not generalize the entire paper into a question. Instead, carefully select the most relevant and specific part of the paper (e.g., a key finding, methodological detail, or discussion point) and craft a detailed question around it. The question should be highly specific and require a nuanced understanding of the paper to answer. Avoid generating general or overly simplistic questions.
6. **Content Relevance and Accuracy**
Ensure that the questions are directly derived from the content of the paper. Avoid generating questions that are irrelevant or based on incorrect facts. The questions should accurately reflect the findings, methodologies, or discussions presented in the paper.
7. **Independent and Complete Question, Answer, and Explanation**
- **Self-contained**: The question, answer, and explanation must be self-contained and complete. Do not use phrases such as "this article" "this paper" "according to the article" "according to the paper" or similar references to the paper.
- **Avoid Personal Pronouns**: Additionally, avoid using personal pronouns like "we" or "our." The question, answer, and explanation must stand alone and be understandable without additional context.
- **Avoid Analysis of the Paper**: Do not include any analysis of the paper in the questions, answers, or explanations. Instead, transform the analysis into independent statements that are self-sufficient and do not rely on the paper for context.
**Example**
{
"question": "The generated question based on the paper.",
"answer": "The corresponding answer to the question.",
"explanation": [
"Step 1: Explanation for the first step.",
"Step 2: Explanation for the second step.",
"Step 3: Explanation for the third step."

---

```
        ]
    }
```

Figure 9: **Complete QA Construction Prompt.**

After constructing the QA pair, we will also refine the QA. The main purpose is to remove extra references that may exist in the QA, such as mentioning papers or non-existent figures in the question. The refine prompt is as follows.

---

**QA Refine Prompt**

**System Prompt**
Please modify the following question, answer, and explanation to remove any expressions related to "paper," "article," "study," or similar references. Ensure that the question is complete and can be answered directly based on the explanation provided, without requiring any additional context or knowledge of the paper. After modification, maintain the original dictionary format and ensure that the meaning of the questions, answers, and explanations remains unchanged.
**Instructions**
1. **Remove References to the Paper**:
- Eliminate any phrases such as "this paper," "the article," "according to the study," or similar references.
- Ensure the question, answer, and explanation are self-contained and do not rely on external sources for understanding.
2. **Maintain Clarity and Completeness**:
- The question should be clear, specific, and able to stand alone, without referencing the paper.
- The answer should directly address the question without referencing the paper.
- The explanation should provide sufficient detail to justify the answer, using independent and self-sufficient statements, without referencing the paper.
3. **Preserve the Dictionary Format**:
- Keep the output in the original dictionary format, including the keys 'question', 'answer', and 'explanation'.
- Ensure that the meaning of the questions, answers, and explanations remains unchanged.

---

Figure 10: **QA Refine Prompt.**

## A.3 SCIENTIFIC EXPLORATION DIALOGUE CONSTRUCTION

The Earth-Gold dataset construction from the high-citation paper collection $P_{hc}$ involves a two-phase processing pipeline:

1. **Paper Structuring Phase**: Decompose each paper into four core components:
   - Summary of existing methods ($M_0$)
   - Limitations of existing methods ($L_0$)
   - Proposed new methods ($M_1$)
   - Potential limitations of new methods ($L_1$)
2. **Dialogue Generation Phase**: Generate two-turn scientific exploration dialogues

The paper structuring prompt is presented as follows:

---

**Paper Structuring Prompt**

**System Prompt**

You are an AI research assistant with expertise in analyzing and structuring academic papers. Your task is to extract and organize the content of a research paper into five specific sections: research direction, methods of previous work, limitations of previous work, method of this work, and limitation of this work. Follow the instructions carefully and output only a dictionary (dict) as specified.

**Instructions**

1. **Task Description**:
- Extract and categorize the content of the provided research paper into the following five sections:
1) **Research direction**
2) **Methods of previous work**
3) **Limitations of previous work**
4) **Method of this work**
5) **Limitation of this work**
- Exclude all other sections or content not related to these five categories.
2. **Output Format**:
- The output must be a Python dictionary (dict) with the following structure:
{
"research direction": "xxx",
"methods of previous work": "xxx",
"limitations of previous work": "xxx",
"method of this work": "xxx",
"limitation of this work": "xxx"
}
- Replace '"xxx"' with the extracted content for each section.
3. **Extraction Rules**:
- **research direction**: Identify the primary focus, goals, or areas of investigation in the paper. Avoid using terms such as "paper," "study," "work," or "thesis." Instead, directly state the research direction in a concise and general manner.
- **methods of previous work**: Summarize the methodologies or approaches used in prior research relevant to the paper.
- **limitations of previous work**: Highlight the shortcomings, gaps, or challenges in previous research.
- **method of this work**: Describe the methodology or approach proposed or used in the current paper.
- **limitation of this work**: Identify and summarize any limitations or weaknesses in the current work's approach.
4. **Important Notes**:
- Strictly adhere to the five sections outlined above.
- Do not include any additional text, explanations, or commentary outside the dictionary.
- Ensure the output is concise, clear, and directly relevant to the specified sections.
- Elaborate on the content with as much detail as possible, retaining specific numerical values, dataset names, method names, author names, etc., and ensure the content is highly professional and information-rich.

**Example**

{
"research direction": "xxx",
"methods of previous work": "xxx",
"limitations of previous work": "xxx",
"method of this work": "xxx",
"limitation of this work": "xxx"
}

Figure 11: **Paper Structuring Prompt.**

The structured paper content enables the generation of two progressive dialogue rounds:

- **Round 1**: The LLM summarizes existing methods ($M_0$) and analyzes their limitations ($L_0$). This round evaluates the model's understanding of domain-specific research methodologies.

- **Round 2**: Building upon Round 1, the LLM proposes innovative methods ($M_1$) while reflecting on potential limitations ($L_1$). This round assesses the model's scientific innovation capability and critical reflection skills.

The dialogue construction prompt is presented as follows:

---

**Dialogue Construction Prompt**

**System Prompt**
You are an AI research assistant specializing in refining supervised fine-tuning (SFT) data for large language models. Your task is to enhance the quality and linguistic diversity of the provided SFT data while preserving its original structure and content.
**Instructions**
1. **Task Description**:
- Polish the text in the SFT data, including both the "user" and "assistant" parts.
- Ensure the polished text is clear, concise, and linguistically diverse while maintaining the original meaning and intent.
2. **Refinement Requirements**:
- Articulate the logic with utmost clarity, employing logical conjunctions to underscore logical relationships where pertinent.
- Elaborate on the content with as much detail as possible, retaining specific numerical values, dataset names, method names, author names, etc., and ensure the content is highly professional and information-rich.
- While retaining the original meaning of the sentence, appropriately add or modify it to make the unsmooth conversation content smooth.
3. **Constraints**:
- Do not add, remove, or alter the number of dialogue turns.
- Preserve the original structure and role labels ("user" and "assistant").
- Use a variety of linguistic styles, vocabulary, and phrasing to increase the diversity of the text.
4. **Output Requirements**:
- Output the modified SFT data as a dialogue list in the same JSON format as the original.
- Ensure the polished text is natural, engaging, and suitable for training large language models.
**Example**
[
{"role": "user", "content": "<INPUT1> Could you provide an overview of related work and discuss their key limitations?"},
{"role": "assistant", "content": "Certainly. The related works include the following. <INPUT2> However, these approaches face several limitations. <INPUT3>"},
{"role": "user", "content": "Given these existing works and their limitations, can you propose a new method and evaluate its potential drawbacks?"},
{"role": "assistant", "content": "Building on these foundations, the proposed method is structured as follows. <INPUT4> Despite its advantages, this method has certain limitations. <INPUT5>"}
]

---

Figure 12: **Dialogue Construction Prompt.**

The <INPUT1> to <INPUT5> in the prompt are replaced by the values in the dictionary obtained in paper structuring.

### A.4 Expert Annotation Guidelines

To ensure that all benchmark questions possess scientific value, specificity, and challenge, we recruited experts with Master's or PhD degrees in Earth sciences (e.g., meteorology, geology) to evaluate each candidate question. Experts scored each question along six dimensions:

1. **Scientific Method**: Does the question involve concrete scientific methods such as data processing, analysis, or experiments?

2. **Data Source**: Does the question include specific datasets or experimental data?

3. **Temporal or Spatial Context**: Does the question reference a specific time or location, such as an earthquake, typhoon, or drought event?

4. **Literature Citation**: Does the question reference a specific scientific paper?

5. **Method Details**: If a new method is proposed, is it described in sufficient detail (step-by-step procedures, method names, etc.)?

6. **Limitation Analysis**: Does the question include a logical, in-depth discussion of method limitations?

Each dimension was scored on a scale of 0 to 2 (0 = not present, 1 = present but insufficiently detailed, 2 = very specific). We retained only those questions for which at least three dimensions received a score greater than 0 and the total score was at least 5. This scoring system ensures that retained questions are scientifically valuable, specific, and challenging.

Table 8: Expert Evaluation Dimensions and Scoring Criteria

| Dimension | Description | Score Range |
|---|---|---|
| Scientific Method | Does the question involve data analysis or experiments? | 0–2 |
| Data Source | Are datasets or sources explicitly specified? | 0–2 |
| Temporal / Spatial Context | Does it reference a specific scientific event or location? | 0–2 |
| Literature Citation | Is a scientific paper cited? | 0–2 |
| Method Details | Are methods described in steps or named specifically? | 0–2 |
| Limitation Analysis | Is there an in-depth discussion of limitations? | 0–2 |

### A.5 LLM Data Construction and Evaluation: Biases and Limitations

Although there is some bias in using LLM for data construction (Skarlinski et al., 2024) and evaluation (Ye et al., 2024), we will try to reduce this bias as much as possible from the following three perspectives.

**Reconstruction-based Generation Instead of Free Creation.** The construction process relies on **reconstruction from original papers** rather than free-form generation. The LLM extracts and reorganizes content from the source material, ensuring scientific accuracy and factual fidelity. In this context, all outputs in *Earth-Gold* are fully manually verified. For *Earth-Iron* and *Earth-Silver*, a combination of model difficulty calibration and human inspection ensures quality control. Specifically, during difficulty calibration, we let LLMs attempt the questions: besides filtering out questions that most models answer correctly, this process identifies questions that all models answer incorrectly. Such questions may either be genuinely difficult or flawed; therefore, human experts review and remove questions that contain errors, while retaining those that are correctly constructed but challenging. All remaining questions have at least one model successfully answering them, ensuring that the question construction is reliable and scientifically sound.

**Fine-grained Structured Construction Process.** When structuring the papers, we consider their **sectional organization** and employ rule-based parsing combined with LLM refinement to efficiently and accurately structure the documents. Specifically, MinerU is used to parse PDF papers and output Markdown format. Using Markdown syntax, we extract first- and second-level headings and assign content to common sections such as *abstract*, *introduction*, *related work*, *method*, *experiment*, *discussion* (optional), and *conclusion*. Papers that do not fit these rules are manually structured

by human experts. Subsequently, we apply LLM refinement to produce the four parts defined in Section 3.4, namely $(M^0, L^0, M^1, L^1)$, where $(M^0, L^0)$ (existing work) is primarily extracted from *related work*, and $(M^1, L^1)$ (new work) is derived from *method*, *experiment*, and *discussion* sections. Summarizing sections such as *abstract*, *introduction*, and *conclusion* are provided as context, ensuring that the model captures the overall research direction while focusing on specific sections for accurate summarization.

**Contrast-based Evaluation Ensures Relative Objectivity and Consistency.** In evaluation, the computation of **win rate** and **retention rate** uses LLMs. For free-response questions, win rate is calculated by comparing model responses against reference answers. Retention rate is derived from ranking the responses relative to the reference answers. Importantly, both metrics rely on **comparative ranking** rather than direct LLM scoring, which can introduce bias and is difficult to quantify. Comparative ranking provides a relatively objective and consistent assessment across different models.

## A.6 QA EXAMPLES

---

**QA Example 1 (Multiple Choice)**

**Question**
What factor plays a pivotal role in the formation of sapropel S1 in the Mediterranean, and what evidence suggests its influence during the last interglacial insolation maximum?
A) Increased wind stress and evidence from high sedimentation rates.
B) Monsoon-fueled freshwater input from the Nile River and synchronized timing of sapropel S1 deposition with wetter conditions in North Africa.
C) Enhanced salinity from increased evaporation and evidence from salinity isotope ratios.
D) Elevated ocean temperatures and evidence from fossil coral records.

**Explanation**
Monsoon-fueled freshwater input, particularly from the Nile River, significantly affected sapropel S1 formation.
The timing of sapropel S1 deposition in the eastern Mediterranean closely coincided with periods of increased Nile River discharge due to enhanced monsoon strength over North Africa.
Evidence suggests these monsoonal shifts resulted in wetter conditions, which aligned with the insolation maxima during the last interglacial, promoting the deposition of organic-rich sapropels.

**Answer**
B

---

Figure 13: **QA Example 1.**

---

**QA Example 2 (True or False)**

**Question**
Atmospheric humidity in Nanjing's urban canopy layer demonstrates significant nighttime differences among local climate zones, particularly due to moisture deficits during colder months, and shows clear seasonal patterns in humidity ratios with greater discrepancies in warmer months than colder months.

**Explanation**
Step 1: Significant differences in atmospheric humidity among local climate zones are observed during nighttime, highlighting variations in humidity ratios across seasons.
Step 2: Negative humidity ratio values (moisture deficits) are more frequent during colder months, while positive values (moisture excess) are noted more often in warmer months.
Step 3: Seasonal patterns are more pronounced in humidity ratio differences, showing greater discrepancies during warmer months, supporting the statement's claim about seasonal variations.

---

> **Answer**
> True

Figure 14: **QA Example 2.**

---

**QA Example 3 (Fill in the Blanks)**

**Question**
In the comparison between SP-CCSM4 and CCSM4, the difference in projected ENSO-shear relationships is attributed to the varying intensity and spatial extent of anomalous westerlies at upper levels during El NINO events. Specifically, while the climatology of easterly flow at _____ simulated in CCSM4 is weaker and retreated eastward compared to SP-CCSM4, the projected westerly anomalies due to ENSO are _____ in CCSM4 relative to those in SP-CCSM4.

**Explanation**
Step 1: The climatology of easterly flow at 200 hPa is noted to be weaker and shifted eastward in CCSM4 compared to SP-CCSM4.
Step 2: During El NINO events, CCSM4 projects stronger westerly anomalies than SP-CCSM4 at upper atmospheric levels, contributing to variability in ENSO-shear relationships.
Step 3: These differences in the intensity and spatial distribution of westerly anomalies directly influence the projected ENSO-shear relationships, highlighting the distinct behavior between the two models in response to ENSO in a warmer climate.

**Answer**
200 hpa, stronger

Figure 15: **QA Example 3.**

---

**QA Example 4 (Free QA)**

**Question**
How does the redistribution of aerosol particles captured through nucleation and impaction scavenging into various drop size categories affect the dynamics and ultimate removal process of aerosol particles in the cloud environment, and what implications do the stochastic collision-coalescence process and drop growth mechanisms have on the distribution and contamination levels observed in precipitation-sized drops?

**Explanation**
Step 1: Aerosol particles are captured in cloud drops through nucleation and impaction scavenging, with their subsequent redistribution depending on the stochastic collision-coalescence processes of drop interactions.
Step 2: As smaller drops merge and grow, aerosol particle mass becomes concentrated in larger precipitation-sized drops, which are more effective in removal due to their increased mass and falling speed.
Step 3: The stochastic collision-coalescence process also results in varying contamination levels, with smaller drops being highly contaminated due to their increased aerosol-to-water mass ratios, while larger drops ensure bulk removal of aerosol mass, significantly affecting the efficiency of wet deposition mechanisms in clouds.

**Answer**
The redistribution of aerosol particles captured via nucleation and impaction scavenging significantly alters the dynamics of aerosol removal by moving the aerosol mass into larger drop size categories, thereby concentrating aerosol particles in precipitation-sized drops. During stochastic collision-coalescence processes, smaller cloud drops merge and grow, carrying the aerosol particle mass with them, leading to a coupling between water mass and aerosol mass growth. This results in a redistribution where the main aerosol mass shifts to drops of precipitation size, which are more effective at removing particles from the cloud

environment. Additionally, smaller drops within the cloud exhibit higher contamination levels due to their higher aerosol-to-water mass ratio, while larger precipitation drops accumulate the majority of aerosol mass, thereby ensuring efficient removal from the atmosphere.

Figure 16: **QA Example 4.**

## A.7 EXAMPLES OF SCIENTIFIC EXPLORATION DIALOGUES

Earth-Gold: Scientific Exploration Dialogue Example 1

**User**
I am interested in the study of sedimentary cycles and their depositional rates over extensive geological time spans, with a particular emphasis on how the likelihood of sedimentary rock destruction changes with age. Could you provide a comprehensive overview of the existing literature in this area, highlighting the principal limitations of these studies?

**Assistant**
Absolutely. The primary works in this domain include Gregor's seminal approach, where he aimed to estimate the global denudation rates across various geological epochs. He derived depositional rates by analyzing the mass-age distribution of current sediments. Nonetheless, these methodologies encounter certain restrictions. For instance, Gregor himself cast doubt on his findings regarding depositional rates during the pre-Carboniferous phases of the Phanerozoic eon. His assumption of a uniform probability of destruction for all sedimentary rocks, regardless of their age, failed to consider fluctuations in the half-lives of rocks over time.

**User**
Considering these existing studies and their constraints, could you suggest an innovative approach and discuss its potential drawbacks?

**Assistant**
Leveraging the insights from previous studies, I propose a novel approach which follows this outline: recharting Gregor's data using semilogarithmic graphs, converting the data into units of grams per year, and fitting these values as exponential functions. This would be complemented by modeling sedimentary cycles that exhibit different half-lives to account for mass-age relationships. Despite its innovative aspects, this approach is not without limitations. While it successfully elucidates Gregor's mass-age associations, it lacks uniqueness and posits a steady total sedimentary mass across cycles, which might overlook some of the more intricate geological phenomena.

Figure 17: **Scientific Exploration Dialogue Example 1.**

Earth-Gold: Scientific Exploration Dialogue Example 2

**User**
Please examine the interplay and responsiveness of the convective boundary layer (CBL) under diverse atmospheric and oceanic conditions, including sea surface temperature (SST), surface wind speed, and tropospheric moisture levels. Could you provide a summary of existing research and highlight their main shortcomings?

**Assistant**
Certainly. Related studies include the following. Betts (1973) introduced a closure model for the subcloud layer that utilized an adapted subsidence approach and energy balance closures consistent with Sarachik's tropical model (1978). Nonetheless, these methods exhibit several limitations. For instance, models by Betts and Sarachik were based on assumptions that might not thoroughly represent the coupling dynamics between the CBL and varying atmospheric parameters.

**User**
Considering these previous studies and their limitations, could you propose a novel approach and assess its potential limitations?
**Assistant**
Drawing inspiration from these constructs, the proposed approach is defined as follows. This work formulates a detailed model to analyze the CBL equilibrium by evaluating uncoupled solutions influenced by SST, wind speed, and tropospheric variations, along with assessments of coupled troposphere solutions and their responsiveness to changes. Despite its benefits, this method does contain some limitations. The model might encounter challenges in accurately forecasting the CBL's behavior under extreme climatic conditions or when integrating intricate feedback mechanisms not entirely accounted for in the theoretical framework.

Figure 18: **Scientific Exploration Dialogue Example 2.**

## A.8 USAGE OF LANGUAGE MODELS

We utilized a large language model (LLM) to aid in the preparation of this manuscript. Its use was limited to editorial tasks, including proofreading for typographical errors, correcting grammar, and improving the clarity and readability of the text.

