# Journal List

Below is a list of high-quality journals from the hierarchical collection of paper data.

| Name | ShortName | ISSN |
| --- | --- | --- |
| CELL | CELL | 0092-8674 |
| NATURE | NATURE | 0028-0836 |
| SCIENCE | SCIENCE | 0036-8075 |
| NATURE BIOTECHNOLOGY | NAT BIOTECHNOL | 1087-0156 |
| NATURE CELL BIOLOGY | NAT CELL BIOL | 1465-7392 |
| NATURE CHEMICAL BIOLOGY | NAT CHEM BIOL | 1552-4450 |
| NATURE CHEMISTRY | NAT CHEM | 1755-4330 |
| NATURE CLIMATE CHANGE | NAT CLIM CHANGE | 1758-678X |
| NATURE COMMUNICATIONS | NAT COMMUN | 2041-1723 |
| NATURE DIGEST | NAT DIGEST | 1880-0556 |
| NATURE GENETICS | NAT GENET | 1061-4036 |
| NATURE GEOSCIENCE | NAT GEOSCI | 1752-0894 |
| NATURE IMMUNOLOGY | NAT IMMUNOL | 1529-2908 |
| NATURE MATERIALS | NAT MATER | 1476-1122 |
| NATURE MEDICINE | NAT MED | 1078-8956 |
| NATURE METHODS | NAT METHODS | 1548-7091 |
| NATURE NANOTECHNOLOGY | NAT NANOTECHNOL | 1748-3387 |
| NATURE NEUROSCIENCE | NAT NEUROSCI | 1097-6256 |
| NATURE PHOTONICS | NAT PHOTONICS | 1749-4885 |
| NATURE PHYSICS | NAT PHYS | 1745-2473 |
| NATURE REVIEWS CANCER | NAT REV CANCER | 1474-175X |
| NATURE REVIEWS CARDIOLOGY | NAT REV CARDIOL | 1759-5002 |
| NATURE REVIEWS CLINICAL ONCOLOGY | NAT REV CLIN ONCOL | 1759-4774 |
| NATURE REVIEWS DRUG DISCOVERY | NAT REV DRUG DISCOV | 1474-1776 |
| NATURE REVIEWS ENDOCRINOLOGY | NAT REV ENDOCRINOL | 1759-5029 |
| NATURE REVIEWS GASTROENTEROLOGY & HEPATOLOGY | NAT REV GASTRO HEPAT | 1759-5045 |
| NATURE REVIEWS GENETICS | NAT REV GENET | 1471-0056 |
| NATURE REVIEWS IMMUNOLOGY | NAT REV IMMUNOL | 1474-1733 |
| NATURE REVIEWS MICROBIOLOGY | NAT REV MICROBIOL | 1740-1526 |
| NATURE REVIEWS MOLECULAR CELL BIOLOGY | NAT REV MOL CELL BIO | 1471-0072 |
| NATURE REVIEWS NEPHROLOGY | NAT REV NEPHROL | 1759-5061 |
| NATURE REVIEWS NEUROLOGY | NAT REV NEUROL | 1759-4758 |
| NATURE REVIEWS NEUROSCIENCE | NAT REV NEUROSCI | 1471-003X |
| NATURE REVIEWS RHEUMATOLOGY | NAT REV RHEUMATOL | 1759-4790 |
| NATURE REVIEWS UROLOGY | NAT REV UROL | 1759-4812 |
| NATURE STRUCTURAL & MOLECULAR BIOLOGY | NAT STRUCT MOL BIOL | 1545-9993 |
| ACCOUNTS OF CHEMICAL RESEARCH | ACCOUNTS CHEM RES | 0001-4842 |
| ADVANCES IN PHYSICS | ADV PHYS | 0001-8732 |
| ANNUAL REVIEW OF ASTRONOMY AND ASTROPHYSICS | ANNU REV ASTRON ASTR | 0066-4146 |

| | | |
|---|---|---|
| ANNUAL REVIEW OF BIOCHEMISTRY | ANNU REV BIOCHEM | 0066-4154 |
| ANNUAL REVIEW OF IMMUNOLOGY | ANNU REV IMMUNOL | 0732-0582 |
| ANNUAL REVIEW OF NEUROSCIENCE | ANNU REV NEUROSCI | 0147-006X |
| ANNUAL REVIEW OF PATHOLOGY MECHANISMS OF DISEASE | ANNU REV PATHOL-MECH | 1553-4006 |
| ANNUAL REVIEW OF PHARMACOLOGY AND THXICOLOGY | ANNU REV PHARMACOL | 0362-1642 |
| ANNUAL REVIEW OF PLANT BIOLOGY | ANNU REV PLANT BIOL | 1543-5008 |
| CA-A CANCER JOURNAL FOR CLINICIANS | CA-CANCER J CLIN | 0007-9235 |
| CANCER CELL | CANCER CELL | 1535-6108 |
| CELL STEM CELL | CELL STEM CELL | 1934-5909 |
| CHEMICAL REVIEWS | CHEM REV | 0009-2665 |
| CHEMICAL SOCIETY REVIEWS | CHEM SOC REV | 0306-0012 |
| IEEE ENG MED BIOL | IEEE ENG MED BIOL | 0739-5175 |
| JAMA-JOURNAL OF THE AMERICAN MEDICAL ASSOCIATION | JAMA-J AM MED ASSOC | 0098-7484 |
| LANCET | LANCET | 0140-6736 |
| LANCET NEUROLOGY | LANCET NEUROL | 1474-4422 |
| LANCET ONCOLOGY | LANCET ONCOL | 1470-2045 |
| LIVING REVIEWS IN RELATIVITY | LIVING REV RELATIV | 1433-8351 |
| NEW ENGLAND JOURNAL OF MEDICINE | NEW ENGL J MED | 0028-4793 |
| PHARMACOLOGICAL REVIEWS | PHARMACOL REV | 0031-6997 |
| PHYSICS REPORTS-REVIEW SECTION OF PHYSICS LETTERS | PHYS REP | 0370-1573 |
| PHYSIOLOGICAL REVIEWS | PHYSIOL REV | 0031-9333 |
| PROGRESS IN MATERIALS SCIENCE | PROG MATER SCI | 0079-6425 |
| PROGRESS IN POLYMER SCIENCE | PROG POLYM SCI | 0079-6700 |
| REVIEWS OF MODERN PHYSICS | REV MOD PHYS | 0034-6861 |
| ANGEWANDTE CHEMIE-INTERNATIONAL EDITION | ANGEW CHEM INT EDIT | 1433-7851 |
| JOURNAL OF THE AMERICAN CHEMICAL SOCIETY | J AM CHEM SOC | 0002-7863 |
| PHYSICAL REVIEW LETTERS | PHYS REV LETT | 0031-9007 |
| PROCEEDINGS OF THE NATIONAL ACADEMY OF SCIENCES OF THE UNITED STATES OF AMERICA | P NATL ACAD SCI USA | 0027-8424 |
| AAPG BULLETIN | AAPG BULL | 0149-1423 |
| ACM TRANSACTIONS ON GRAPHICS | ACM T GRAPHIC | 0730-0301 |
| ACM TRANSACTIONS ON SOFTWARE ENGINEERING METHODOLOGY | ACM T SOFTW ENG METH | 1049-331X |
| ACS APPLIED MATERIALS AND INTERFACES | ACS APPL MATER INTER | 1944-8244 |
| ACS CHEMICAL BIOLOGY | ACS CHEM BIOL | 1554-8929 |
| ACS NANO | ACS NANO | 1936-0851 |
| ACTA MATERILIA | ACTA MATER | 1359-6454 |
| ACTA MATHEMATICA | ACTA MATH | 0001-5962 |
| ADVANCED ENERGY MATERIALS | ADV ENERGY MATER | 1614-6832 |
| ADVANCED FUNCTIONAL MATERIALS | ADV FUNCT MATER | 1616-301X |
| ADVANCED MATERIALS | ADV MATER | 0935-9648 |
| AEROSOL SCIENCE AND TECHNOLOGY | AEROSOL SCI TECH | 0278-6826 |
| AGRICULTURAL AND FOREST METEOROLOGY | AGR FOREST METEOROL | 0168-1923 |
| AICHE JOURNAL | AICHE J | 0001-1541 |
| AMERICAL JOURNAL OF PATHOLOGY | J PATHOL | 0022-3417 |
| AMERICAN JOURNAL OF CLINICAL NUTRITION | AM J CLIN NUTR | 0002-9165 |

| AMERICAN JOURNAL OF HUMAN GENETICS | AM J HUM GENET | 0002-9297 |
|---|---|---|
| AMERICAN JOURNAL OF RESPIRATORY AND CRITICAL CARE MEDICINE | AM J RESP CRIT CARE | 1073-449X |
| AMERICAN JOURNAL OF TRANSPLANTATION | AM J TRANSPLANT | 1600-6135 |
| ANALYTICAL CHEMISTRY | ANAL CHEM | 0003-2700 |
| ANIMAL BEHAVIOUR | ANIM BEHAV | 0003-3472 |
| ANNALS OF EMERGENCY MEDICINE | ANN EMERG MED | 0196-0644 |
| ANNALS OF MATHEMATICS | ANN MATH | 0003-486X |
| ANNALS OF ROBABILITY | ANN PROBAB | 0091-1798 |
| ANNALS OF STATISTICS | ANN STAT | 0090-5364 |
| ANNALS OF SURGERY | ANN SURG | 0003-4932 |
| ANNALS OF THE RHEUMATIC DISEASES | ANN RHEUM DIS | 0003-4967 |
| ANNUAL REVIER OF EARTH AND PLANETARY SCIENCES | ANNU REV EARTH PL SC | 0084-6597 |
| ANNUAL REVIEW OF MARINE SCIENCE | ANNU REV MAR SCI | 1941-1405 |
| APPLIED CATALYSIS B-ENVIRONMENTAL | APPL CATAL B-ENVIRON | 0926-3373 |
| APPLIED ENERGY | APPL ENERG | 0306-2619 |
| APPLIED PHYSICS LETTERS | APPL PHYS LETT | 0003-6951 |
| ARTHRITIS & RHEUMATOLOGY | ARTHRITIS RHEUM-US | 0004-3591 |
| ARTIFICIAL INTELLIGENCE | ARTIF INTELL | 0004-3702 |
| ASTRONOMY AND ASTROPHYSICS | ASTRON ASTROPHYS | 0004-6361 |
| ASTRONOMY AND ASTROPHYSICS REVIEW | ASTRON ASTROPHYS REV | 0935-4956 |
| ATMOSPHERIC CHEMISTRY AND PHYSICS | ATMOS CHEM PHYS | 1680-7316 |
| ATMOSPHERIC ENVIRONMENT | ATMOS ENVIRON | 1352-2310 |
| BIOINFORMATICS | BIOINFORMATICS | 1367-4803 |
| BIOMACROMOLECULES | BIOMACROMOLECULES | 1525-7797 |
| BIOMATERIALS | BIOMATERIALS | 0142-9612 |
| BIOSENSORS AND BIOELECTRONICS | BIOSENS BIOELECTRON | 0956-5663 |
| BIRTH-ISSUES IN PERINATAL CARE | BIRTH-ISS PERINAT C | 0730-7659 |
| BLOOD | BLOOD | 0006-4971 |
| BRAIN | BRAIN | 0006-8950 |
| BUILDING AND ENVIRONMENT | BUILD ENVIRON | 0360-1323 |
| BULLETIN OF THE AMERICAN METEOROLOGICAL SOCIETY | B AM METEOROL SOC | 0003-0007 |
| CANCER DISCOVERY | CANCER DISCOV | 2159-8274 |
| CELL HOST&MICROBE | CELL HOST MICROBE | 1931-3128 |
| CELL METABOLISM | CELL METAB | 1550-4131 |
| CHEMICAL COMMUNICATIONS | CHEM COMMUN | 1359-7345 |
| CHEMICAL SCIENCE | CHEM SCI | 2041-6520 |
| CHEMISTRY OF MATERIALS | CHEM MATER | 0897-4756 |
| CHEMISTRY-A EUROPEAN JOURNAL | CHEM-EUR J | 0947-6539 |
| CHEMOSPHERE | CHEMOSPHERE | 0045-6535 |
| CIRCULATION | CIRCULATION | 0009-7322 |
| CLIMATE DYNAMICS | CLIM DYNAM | 0930-7575 |
| CLINICAL INFECTIOUS DISEASES | CLIN INFECT DIS | 1058-4838 |
| COMMUNICATIONS IN MATHEMATICAL PHYSICS | COMMUN MATH PHYS | 0010-3616 |
| COMMUNICATIONS IN NONLINEAR SCIENCE AND NUMERICAL SIMULATION | COMMUN NONLINEAR SCI | 1007-5704 |

| | | |
|---|---|---|
| COMMUNICATIONS ON PURE AND APPLIED MATHEMATICS | COMMUN PUR APPL MATH | 0010-3640 |
| CURRENT BIOLOGY | CURR BIOL | 0960-9822 |
| DALTON TRANSACTIONS | DALTON T | 1477-9226 |
| DENTAL MATERIALS | DENT MATER | 0109-5641 |
| DEVELOPMENTAL CELL | DEV CELL | 1534-5807 |
| DUKE MATHEMATICAL JOURNAL | DUKE MATH J | 0012-7094 |
| EAR AND HEARING | EAR HEARING | 0196-0202 |
| EARTH AND PLANETARY SCIENCE LETTERS | EARTH PLANET SC LETT | 0012-821X |
| EARTH-SCIENCE REVIEWS | EARTH-SCI REV | 0012-8252 |
| ECOLOGICAL MONOGRAPHS | ECOL MONOGR | 0012-9615 |
| ECOLOGY | ECOLOGY | 0012-9658 |
| ECOLOGY LETTERS | ECOL LETT | 1461-023X |
| EMBO JOURNAL | EMBO J | 0261-4189 |
| ENERGY | ENERGY | 0360-5442 |
| ENERGY & ENVIRONMNENTAL SCIENCE | ENERG ENVIRON SCI | 1754-5692 |
| ENERGY AND BUILDINGS | ENERG BUILDINGS | 0378-7788 |
| ENERGY CONVERSION AND MANAGEMENT | ENERG CONVERS MANAGE | 0196-8904 |
| ENGINEERING GEOLOGY | ENG GEOL | 0013-7952 |
| ENVIRONMENTAL HEALTH PERSPECTIVE | ENVIRON HEALTH PERSP | 0091-6765 |
| ENVIRONMENTAL INTERNATIONAL | ENVIRON INT | 0160-4120 |
| ENVIRONMENTAL MICROBIOLOGY | ENVIRON MICROBIOL | 1462-2912 |
| ENVIRONMETAL SCIENCE AND TECHNOLOGY | ENVIRON SCI TECHNOL | 0013-936X |
| EPIDEMIOLOGY | EPIDEMIOLOGY | 1044-3983 |
| EUROPEAN PHYSICAL JOURNAL C | EUR PHYS J C | 1434-6044 |
| GASTROENTEROLOGY | GASTROENTEROLOGY | 0016-5085 |
| GENES & DEVELOPMENT | GENE DEV | 0890-9369 |
| GENOME BIOLOGY | GENOME BIOL | 1474-7596 |
| GENOME RESEARCH | GENOME RES | 1088-9051 |
| GEOCHIMICA COSMOCHIMICA ACTA | GEOCHIM COSMOCHIM AC | 0016-7037 |
| GEOLOGY | GEOLOGY | 0091-7613 |
| GEOMETRIC AND FUNCTIONAL ANALYSIS | GEOM FUNCT ANAL | 1016-443X |
| GEOPHYSICAL RESEARCH LETTER | GEOPHYS RES LETT | 0094-8276 |
| GEOSCIENTIFIC MODEL DEVELOPMENT | GEOSCI MODEL DEV | 1991-959X |
| GEOTEXTILE AND GEOMEMBRANES | GEOTEXT GEOMEMBRANES | 0266-1144 |
| GLOBAL AND PLANETARY CHANGE | GLOBAL PLANET CHANGE | 0921-8181 |
| GLOBAL BIOGEOCHEMICAL CYCLS | GLOBAL BIOGEOCHEM CY | 0886-6236 |
| GLOBAL CHANGE BIOLOGY | GLOBAL CHANGE BIOL | 1354-1013 |
| GLOBAL ECOLOGY AND BIOGEOGRAPHY | GLOBAL ECOL BIOGEOGR | 1466-822X |
| GLOBAL ENVIRONMENTAL CHANGE | GLOBAL ENVIRON CHANG | 0959-3780 |
| GUT | GUT | 0017-5749 |
| HUMAN REPRODUCTION | HUM REPROD | 0268-1161 |
| HYDROLOGY AND EARTH SYSTEM SCIENCE | HYDROL EARTH SYST SC | 1027-5606 |
| IEEE ELECTRON DEVICE LETTERS | IEEE ELECTR DEVICE L | 0741-3106 |
| IEEE JOURNAL OF SELECTED TOPICS IN QUANTUM ELECTRONICS | IEEE J SEL TOP QUANT | 1077-260X |

| | | |
|---|---|---|
| IEEE JOURNAL OF SELECTED TOPICS IN SIGNAL PROCESSING | IEEE J-STSP | 1932-4553 |
| IEEE JOURNAL ON SELECTED AREAS IN COMMUNICATIONS | IEEE J SEL AREA COMM | 0733-8716 |
| IEEE TRANS ON PATTERN ANALYSIS AND MACHINE INTELLIGENCE | IEEE T PATTERN ANAL | 0162-8828 |
| IEEE TRANSACTION ON AUTOMATIC CONTROL | IEEE T AUTOMAT CONTR | 0018-9286 |
| IEEE TRANSACTION ON SYATEMS, MAN, AND CYBERNETICS: B | IEEE T SYST MAN CY B | 1083-4419 |
| IEEE TRANSACTIONS CIRCUITS AND SYSTEMS-I | IEEE T CIRCUITS-I | 1549-8328 |
| IEEE TRANSACTIONS ON ANTENNAS AND PROPAGATIONS | IEEE AP | 0018-926X |
| IEEE TRANSACTIONS ON COMPUTERS | IEEE T COMPUT | 0018-9340 |
| IEEE TRANSACTIONS ON GEOSCIENCE AND REMOTE SENSING | IEEE T GEOSCI REMOTE | 0196-2892 |
| IEEE TRANSACTIONS ON IMAGE PROCESSING | IEEE T IMAGE PROCESS | 1057-7149 |
| IEEE TRANSACTIONS ON INFORMATION FORENSICS AND SECURITY | IEEE T INF FOREN SEC | 1556-6013 |
| IEEE TRANSACTIONS ON KNOWLEDGE AND DATA ENGINEERING | IEEE T KNOWL DATA EN | 1041-4347 |
| IEEE TRANSACTIONS ON MEDICAL IMAGING | IEEE T MED IMAGING | 0278-0062 |
| IEEE TRANSACTIONS ON MICROWAVE THEORY AND TECHNIQUES | IEEE T MICROW THEORY | 0018-9480 |
| IEEE TRANSACTIONS ON MOBILE COMPUTING | IEEE T MOBILE COMPUT | 1536-1233 |
| IEEE TRANSACTIONS ON PARALLEL AND DISTRIBUTED SYSTEMS | IEEE T PARALL DISTR | 1045-9219 |
| IEEE TRANSACTIONS ON SIGNAL PROCESSING | IEEE T SIGNAL PROCES | 1053-587X |
| IEEE TRANSACTIONS ON SOFTWARE ENGINEERING | IEEE T SOFTWARE ENG | 0098-5589 |
| IEEE TRANSACTIONS ON VISUALIZATION AND COMPUTER GRAPHICS | IEEE T VIS COMPUT GR | 1077-2626 |
| IEEE WIRELESS COMMUNICATIONS | IEEE WIREL COMMUN | 1536-1284 |
| IEEE/ACM Transactions on Networking | IEEE ACM T NETWORK | 1063-6692 |
| IEEE-ASME TRANSACTIONS ON MECHATRONICS | IEEE-ASME T MECH | 1083-4435 |
| IMMUNITY | IMMUNITY | 1074-7613 |
| INDOOR AIR | INDOOR AIR | 0905-6947 |
| INORGANIC CHEMISTRY | INORG CHEM | 0020-1669 |
| International Journal of Computer Vision | INT J COMPUT VISION | 0920-5691 |
| INVENTIONES MATHEMATICAE | INVENT MATH | 0020-9910 |
| ISME JOURNAL | ISME J | 1751-7362 |
| JNCI-JOURNAL OF THE NATIONAL CANCER INSTITUTE | JNCI-J NATL CANCER I | 0027-8874 |
| JOURNAL OF PETROLOGY | J PETROL | 0022-3530 |
| JOURNAL OF AEROSOL SCIENCE | J AEROSOL SCI | 0021-8502 |
| JOURNAL OF BIOGEOGRAPHER | J BIOGEOGR | 0305-0270 |
| JOURNAL OF BIOLOGICAL CHEMISTRY | J BIOL CHEM | 0021-9258 |
| JOURNAL OF CELL BIOLOGY | J CELL BIOL | 0021-9525 |
| Journal of Climate | J CLIMATE | 0894-8755 |
| JOURNAL OF CLINICAL INVESTIGATION | J CLIN INVEST | 0021-9738 |
| JOURNAL OF CLINICAL ONCOLOGY | J CLIN ONCOL | 0732-183X |
| JOURNAL OF CONTROLLED RELEASE | J CONTROL RELEASE | 0168-3659 |
| JOURNAL OF COSMOLOGY AND ASTROPARTICLE PHYSICS | J COSMOL ASTROPART P | 1475-7516 |
| JOURNAL OF DENTAL RESEARCH | J DENT RES | 0022-0345 |
| JOURNAL OF EXPERIMENTAL MEDICINE | J EXP MED | 0022-1007 |
| Journal of Geophysical Research: Atmosphere | J GEOPHYS RES: Atmosphere | 0148-0227 |
| Journal of Geophysical Research: Oceans | J GEOPHYS RES: Ocean | 0148-0227 |
| JOURNAL OF GEOPHYSICAL RESEARCH-SOLID EARTH | J GEOPHYS RES | 0148-0227 |

| JOURNAL OF HAZARDOUS MATERIALS | J HAZARD MATER | 0304-3894 |
|---|---|---|
| JOURNAL OF HIGH   ENERGY PHYSICS | J HIGH ENERGY PHYS | 1126-6708 |
| JOURNAL OF HYDROLOGY | J HYDROL | 0022-1694 |
| JOURNAL OF INVESTIGATIVE DERMATOLOGY | J INVEST DERMATOL | 0022-202X |
| JOURNAL OF LIGHTWAVE TECHNOLOGY | J LIGHTWAVE TECHNOL | 0733-8724 |
| Journal of Machine Learning Research | J MACH LEARN RES | 1532-4435 |
| JOURNAL OF NEUROSCIENCE | J NEUROSCI | 0270-6474 |
| JOURNAL OF OPERATIONS MANAGEMENT | J OPER MANAG | 0272-6963 |
| JOURNAL OF ORGANIC CHEMISTRY | J ORG CHEM | 0022-3263 |
| JOURNAL OF PEDIATRICS | J PEDIATR-US | 0022-3476 |
| JOURNAL OF PHYSICAL CHEMISTRY LETTERS | J PHYS CHEM LETT | 1948-7185 |
| JOURNAL OF POWER SOURCES | J POWER SOURCES | 0378-7753 |
| Journal of the ACM | J ACM | 0004-5411 |
| JOURNAL OF THE AMERICAN MATHEMATICAL SOCIETY | J AM MATH SOC | 0894-0347 |
| JOURNAL OF THE AMERICAN SOCIETY OF NEPHROLOGY | J AM SOC NEPHROL | 1046-6673 |
| Journal of the Atmospheric Sciences | J ATMOS SCI | 0022-4928 |
| JOURNAL OF THE EUROPEAN CERAMIC SOCIETY | J EUR CERAM SOC | 0955-2219 |
| JOURNAL OF THE EUROPEAN MATHEMATICAL SOCIETY | J EUR MATH SOC | 1435-9855 |
| JOURNAL OF VIROLOGY | J VIROL | 0022-538X |
| KIDNEY INTERNATIONAL | KIDNEY INT | 0085-2538 |
| LASER PHOTONICS REVIEW | LASER PHOTONICS REV | 1863-8880 |
| LEUKEMIA | LEUKEMIA | 0887-6924 |
| LIVING REVIEWS IN SOLAR PHYSICS | LIVING REV SOL PHYS | 1614-4961 |
| MACROMOLECULES | MACROMOLECULES | 0024-9297 |
| MANAGEMENT SCIENCE | MANAGE SCI | 0025-1909 |
| MARINE GEOLOGY | MAR GEOL | 0025-3227 |
| MATERIALS CHARACTERIZATION | MATER CHARACT | 1044-5803 |
| MATERIALS SCIENCE AND ENGINEERING: REPORTS | MAT SCI ENG R | 0927-796X |
| MATHEMATICAL PROGRAMMING | MATH PROGRAM | 0025-5610 |
| MEMOIRS OF THE AMERICAN MATHEMATICAL SOCIETY | MEM AM MATH SOC | 0065-9266 |
| MOLECULAR CELL | MOL CELL | 1097-2765 |
| MONTHLY NOTICES OF THE ROYAL ASTRONOMICAL SOCIETY | MON NOT R ASTRON SOC | 0035-8711 |
| NANO LETTERS | NANO LETT | 1530-6984 |
| NANOTOXICOLOGY | NANOTOXICOLOGY | 1743-5390 |
| NEURON | NEURON | 0896-6273 |
| NEW JOURNAL OF PHYSICS | NEW J PHYS | 1367-2630 |
| OMEGA-INTERNATIONAL JOURNAL OF MANAGEMENT SCIENCE | OMEGA-INT J MANAGE S | 0305-0483 |
| OPERATIONS RESEARCH | OPER RES | 0030-364X |
| OPHTHALMOLOGY | OPHTHALMOLOGY | 0161-6420 |
| OPTICS EXPRESS | OPT EXPRESS | 1094-4087 |
| OPTICS LETTERS | OPT LETT | 0146-9592 |
| ORE GEOLOGY REVIEW | ORE GEOL REV | 0169-1368 |
| ORGANIC LETTERS | ORG LETT | 1523-7060 |
| PAIN | PAIN | 0304-3959 |

| | | |
|---|---|---|
| PALAEOGEOGRAPHY PALAEOCLIMATOLOGY PALAEOECOLOGY | PALAEOGEOGR PALAEOCL | 0031-0182 |
| PHYSICAL REVIEW E | PHYS REV E | 1539-3755 |
| PHYSICAL REVIEW A | PHYS REV A | 1050-2947 |
| PHYSICAL REVIEW B | PHYS REV B | 1098-0121 |
| PHYSICAL REVIEW C | PHYS REV C | 0556-2813 |
| PHYSICAL REVIEW D | PHYS REV D | 1550-7998 |
| PHYSICAL REVIEW X | PHYS REV X | 2160-3308 |
| PLANT CELL | PLANT CELL | 1040-4651 |
| PLANT JOURNAL | PLANT J | 0960-7412 |
| PLANT PHYSIOLOGY | PLANT PHYSIOL | 0032-0889 |
| PLOS BIOLOGY | PLOS BIOL | 1545-7885 |
| PROCEEDINGS OF THE LONDON MATHEMATICAL SOCIETY | P LOND MATH SOC | 0024-6115 |
| PROCEEDINGS OF THE ROYAL SOCIETY B-BIOLOGICAL SCIENCES | P ROY SOC B-BIOL SCI | 0962-8452 |
| QUATERNARY SCIENCE REVIEWS | QUATERNARY SCI REV | 0277-3791 |
| RADIOLOGY | RADIOLOGY | 0033-8419 |
| REMOTE SENSING OF ENVIRONMENT | REMOTE SENS ENVIRON | 0034-4257 |
| RENEWABLE AND SUSTAINABLE ENERGY REVIEWS | RENEW SUST ENERG REV | 1364-0321 |
| SCIENTIFIC REPORTS | SCI REP-UK | 2045-2322 |
| SIAM JOURNAL ON NUMERICAL ANALYSIS | SIAM J NUMER ANAL | 0036-1429 |
| SIAM JOURNAL ON OPTIMIZATION | SIAM J OPTIMIZ | 1052-6234 |
| SOFT MATTER | SOFT MATTER | 1744-683X |
| SOIL BIOLOGY & BIOCHEMISTRY | SOIL BIOL BIOCHEM | 0038-0717 |
| SOLAR ENERGY MATERIALS AND SOLAR CELLS | SOL ENERG MAT SOL C | 0927-0248 |
| SPACE SCIENCE REVIEWS | SPACE SCI REV | 0038-6308 |
| SPINE JOURNAL | SPINE J | 1529-9430 |
| STROKE | STROKE | 0039-2499 |
| TECTONICS | TECTONICS | 0278-7407 |
| THE ASTRONOMICAL JOURNAL | ASTRON J | 0004-6256 |
| THE ASTROPHYSICAL JOURNAL | ASTROPHYS J | 0004-637X |
| THE ASTROPHYSICAL JOURNAL LETTERS | ASTROPHYS J LETT | 2041-8205 |
| THE ASTROPHYSICAL JOURNAL SUPPLEMENT SERIES | ASTROPHYS J SUPPL S | 0067-0049 |
| TOXICOLOGICAL SCIENCE | TOXICOL SCI | 1096-6080 |
| TRANSPORTATION RESEARCH PART B: METHODOLOGICAL | TRANSPORT RES B-METH | 0191-2615 |
| TRANSPORTATION SCIENCE | TRANSPORT SCI | 0041-1655 |
| WATER RESEARCH | WATER RES | 0043-1354 |
| WATER RESOURCE RESEARCH | WATER RESOUR RES | 0043-1397 |
| ACM Transactions on Knowledge Discovery from Data | ACM T KNOWL DISCOV D | 1556-4681 |
| ACM Transactions on Mathematical Software | ACM T MATH SOFTWARE | 0098-3500 |
| ACS CATALYSIS | ACS CATAL | 2155-5435 |
| ACTA BIOMATERIALIA | ACTA BIOMATER | 1742-7061 |
| ADVANCE IN WATER RESOURCES | ADV WATER RESOUR | 0309-1708 |
| ADVANCED SYNTHESIS AND CATALYSIS | ADV SYNTH CATAL | 1615-4150 |
| AGRICUTURAL WATER MANAGEMENT | AGR WATER MANAGE | 0378-3774 |
| AGRICURE ECOSYSTEMS &   ENVIRONMENT | AGR ECOSYST ENVIRON | 0167-8809 |

| ALLERGY | ALLERGY | 0105-4538 |
|---|---|---|
| AMERICAN JOURNAL OF EPIDEMIOLOGY | AM J EPIDEMIOL | 0002-9262 |
| AMERICAN JOURNAL OF GASTROENTEROLOGY | AM J GASTROENTEROL | 0002-9270 |
| AMERICAN JOURNAL OF KIDNEY DISEASES | AM J KIDNEY DIS | 0272-6386 |
| AMERICAN JOURNAL OF OPHTHALMOLOGY | AM J OPHTHALMOL | 0002-9394 |
| AMERICAN JOURNAL OF PATHOLOGY | AM J PATHOL | 0002-9440 |
| AMERICAN JOURNAL OF PHYSIOLOGY-RENAL PHYSIOLOGY | AM J PHYSIOL-RENAL | 1931-857X |
| AMERICAN JOURNAL OF PUBLIC HEALTH | AM J PUBLIC HEALTH | 0090-0036 |
| AMERICAN JOURNAL OF RESPIRATORY CELL AND MOLECULAR BIOLOGY | AM J RESP CELL MOL | 1044-1549 |
| AMERICAN JOURNAL OF SURGERY | AM J SURG | 0002-9610 |
| ANAESTHESIA | ANAESTHESIA | 0003-2409 |
| ANALYST | ANALYST | 0003-2654 |
| ANALYTICA CHIMICA ACTA | ANAL CHIM ACTA | 0003-2670 |
| ANESTHESIOLOGY | ANESTHESIOLOGY | 0003-3022 |
| ANTIOXIDANTS & REDOX SIGNALING | ANTIOXID REDOX SIGN | 1523-0864 |
| APPLIED CATALYSIS A-GENERAL | APPL CATAL A-GEN | 0926-860X |
| APPLIED CLAY SCIENCE | APPL CLAY SCI | 0169-1317 |
| APPLIED MICROBIOLOGY AND BIOTECHNOLOGY | APPL MICROBIOL BIOT | 0175-7598 |
| APPLIED PHYSICS EXPRESS | APPL PHYS EXPRESS | 1882-0778 |
| APPLIED SURFACE SCIENCE | APPL SURF SCI | 0169-4332 |
| APPLIED THERMAL ENGINEERING | APPL THERM ENG | 1359-4311 |
| AQUATIC TOXICOLOGY | AQUAT TOXICOL | 0166-445X |
| ARTHROSCOPY-THE JOURNAL OF ARTHROSCOPIC AND RELATED SURGERY | ARTHROSCOPY | 0749-8063 |
| AUTOMATICA | AUTOMATICA | 0005-1098 |
| AUTOPHAGY | AUTOPHAGY | 1554-8627 |
| BIOCHEMICAL PHARMACOLOGY | BIOCHEM PHARMACOL | 0006-2952 |
| BIOGEOCHEMISTRY | BIOGEOCHEMISTRY | 0168-2563 |
| BIOGEOSCIENCES | BIOGEOSCIENCES | 1726-4170 |
| BIORESOURCE TECHNOLOGY | BIORESOURCE TECHNOL | 0960-8524 |
| BMC BIOLOGY | BMC BIOL | 1741-7007 |
| BRITISH JOURNAL OF CANCER | BRIT J CANCER | 0007-0920 |
| BRITISH JOURNAL OF PHARMACOLOGY | BRIT J PHARMACOL | 0007-1188 |
| CANCER RESEARCH | CANCER RES | 0008-5472 |
| CARBON | CARBON | 0008-6223 |
| CELL DEATH & DISEASE | CELL DEATH DIS | 2041-4889 |
| CELL DEATH AND DIFFERENTIATION | CELL DEATH DIFFER | 1350-9047 |
| CELL RESEARCH | CELL RES | 1001-0602 |
| CERAMICS INTERNATIONAL | CERAM INT | 0272-8842 |
| CHEMCATCHEM | CHEMCATCHEM | 1867-3880 |
| CHEMICAL ENGINEERING JOURNAL | CHEM ENG J | 1385-8947 |
| CHEMICAL GEOLOGY | CHEM GEOL | 0009-2541 |
| CHEMISTRY-AN ASIAN JOURNAL | CHEM-ASIAN J | 1861-4728 |
| CHEMSUSCHEM | CHEMSUSCHEM | 1864-564X |
| CIRCULATION RESEARCH | CIRC RES | 0009-7330 |

| CIRP ANNALS-MANUFACTURING TECHNOLOGY | CIRP ANN-MANUF TECHN | 0007-8506 |
|---|---|---|
| CLEAN-SOIL AIR WATER | CLEAN-SOIL AIR WATER | 1863-0650 |
| Climate of the Past | CLIM PAST | 1814-9324 |
| CLIMATIC CHANGE | CLIMATIC CHANGE | 0165-0009 |
| CLINICAL JOURNAL OF THE AMERICAN SOCIETY OF NEPHROLOGY | CLIN J AM SOC NEPHRO | 1555-9041 |
| COMMUNICATIONS IN PARTIAL DIFFERENTIAL EQUATIONS | COMMUN PART DIFF EQ | 0360-5302 |
| COMPUTER GRAPHICS FORUM | COMPUT GRAPH FORUM | 0167-7055 |
| COMPUTERS & OPERATIONS RESEARCH | COMPUT OPER RES | 0305-0548 |
| CONTRIBUTION OF MINERAL AND PETROLOGY | CONTRIB MINERAL PETR | 0010-7999 |
| CRYSTAL GROWTH AND DESIGN | CRYST GROWTH DES | 1528-7483 |
| Data Mining and Knowledge Discovery | DATA MIN KNOWL DISC | 1384-5810 |
| DECISION SUPPORT SYSTEMS | DECIS SUPPORT SYST | 0167-9236 |
| DESALINATION | DESALINATION | 0011-9164 |
| DEVELOPMENT | DEVELOPMENT | 0950-1991 |
| DIABETOLOGIA | DIABETOLOGIA | 0012-186X |
| DIASTER ADVANCE | DISASTER ADV | 0974-262X |
| DISCRETE CONTINUOUS DYNAMICAL SYSTEM A | DISCRETE    CONTIN DYN SYST | 1078-0947 |
| DYES AND PIGMENTS | DYES PIGMENTS | 0143-7208 |
| ECOHYDROLOGY | ECOHYDROLOGY | 1936-0584 |
| ELECTROCHEMISTRY COMMUNICATIONS | ELECTROCHEM COMMUN | 1388-2481 |
| EMBO REPORTS | EMBO REP | 1469-221X |
| EMERGENCIAS | EMERGENCIAS | 1137-6821 |
| EMERGING INFECTIOUS DISEASES | EMERG INFECT DIS | 1080-6040 |
| ENDOCRINOLOGY | ENDOCRINOLOGY | 0013-7227 |
| ENDOSCOPY | ENDOSCOPY | 0013-726X |
| ENVIRONMENTAL MODEELING & SOFTWARE | ENVIRON MODELL SOFTW | 1364-8152 |
| ENVIRONMENTAL POLLUTION | ENVIRON POLLUT | 0269-7491 |
| ENVIRONMENTAL RESEARCH | ENVIRON RES | 0013-9351 |
| Environmental Research Letters | ENVIRON RES LETT | 1748-9326 |
| ENVIRONMENTAL SCIENCE & POLICY | ENVIRON SCI POLICY | 1462-9011 |
| ERGODIC THEORY AND DYNAMICAL SYSTEMS | ERGOD THEOR DYN SYST | 0143-3857 |
| EUROPEAN JOURNAL OF OPERATIONAL RESEARCH | EUR J OPER RES | 0377-2217 |
| EUROPEAN PHYSICAL JOURNAL   H | EUR PHYS J H | 2102-6459 |
| EUROPENAN JOURNAL OF FOREST RESARCH | EUR J FOREST RES | 1612-4669 |
| Evolutionary Computation | EVOL COMPUT | 1063-6560 |
| EXPERIMENTAL NEUROLOGY | EXP NEUROL | 0014-4886 |
| EXPERIMENTAL THERMAL AND FLUID SCIENCE | EXP THERM FLUID SCI | 0894-1777 |
| FASEB JOURNAL | FASEB J | 0892-6638 |
| FERTILITY AND STERILITY | FERTIL STERIL | 0015-0282 |
| FOREST AND ECOLOGY MANAGEMENT | FOREST ECOL MANAG | 0378-1127 |
| FREE RADICAL BIOLOGY AND MEDICINE | FREE RADICAL BIO MED | 0891-5849 |
| GEOFLUIDS | GEOFLUIDS | 1468-8115 |
| GEOLOGICAL SOCIETY OF AMERICA BULLETIN | GEOL SOC AM BULL | 0016-7606 |
| GEOTECHNIQUE | GEOTECHNIQUE | 0016-8505 |

| GONDWANA RESEARCH | GONDWANA RES | 1342-937X |
|---|---|---|
| GREEN CHEMISTRY | GREEN CHEM | 1463-9262 |
| HAEMATOLOGICA-THE HEMATOLOGY JOURNAL | HAEMATOL-HEMATOL J | 0390-6078 |
| HEAD AND NECK-JOURNAL FOR THE SCIENCES AND SPECIALTIES OF THE HEAD AND NECK | HEAD NECK-J SCI SPEC | 1043-3074 |
| HEART | HEART | 1355-6037 |
| HOLOCENE | HOLOCENE | 0959-6836 |
| Human Computer Interaction | HUM-COMPUT INTERACT | 0737-0024 |
| HUMUN MOLECULAR GENETICS | HUM MOL GENET | 0964-6906 |
| HYDROLOGICAL PROCESS | HYDROL PROCESS | 1099-1085 |
| IEEE JOURNAL OF SELECTED TOPICS ON APPLIED REMOTE SENSING | IEEE J-STARS | 1939-1404 |
| IEEE PHOTONICS JOURNAL | IEEE PHOTONICS J | 1943-0655 |
| IEEE PHOTONICS TECHNOLOGY LETTERS | IEEE PHOTONIC TECH L | 1041-1135 |
| IEEE TRANSACTION ON   FUZZY SYSTEMS | IEEE T FUZZY SYST | 1063-6706 |
| IEEE TRANSACTION ON NEURAL NETWORKS AND LEARNING SYSTEMS | IEEE T NEUR NET LEAR | 2162-237X |
| IEEE TRANSACTION ON POWER SYSTEM | IEEE T POWER SYST | 0885-8950 |
| IEEE TRANSACTION ON ROBOTICS | IEEE T ROBOT | 1552-3098 |
| IEEE TRANSACTION ON SYATEMS, MAN, AND CYBERNETICS: A | IEEE T SYST MAN CY A | 1083-4427 |
| IEEE TRANSACTION ON SYATEMS, MAN, AND CYBERNETICS: C | IEEE T SYST MAN CY C | 1094-6977 |
| IEEE TRANSACTIONS ON AEROSPACE AND ELECTRONIC SYSTEMS | IEEE T AERO ELEC SYS | 0018-9251 |
| IEEE TRANSACTIONS ON BIOMEDICAL CIRCUITS AND SYSTEMS | IEEE T BIOMED CIRC S | 1932-4545 |
| IEEE TRANSACTIONS ON COMMUNICATIONS | IEEE T COMMUN | 0090-6778 |
| IEEE TRANSACTIONS ON ELECTRON DEVICES | IEEE T ELECTRON DEV | 0018-9383 |
| IEEE Transactions on Evolutionary Computation | IEEE T EVOLUT COMPUT | 1089-778X |
| IEEE TRANSACTIONS ON INDUSTRIAL INFORMATICS | IEEE T IND INFORM | 1551-3203 |
| IEEE Transactions on Multimedia | IEEE T MULTIMEDIA | 1520-9210 |
| IEEE Transactions on Service Computing | IEEE T SERV COMPUT | 1939-1374 |
| IEEE TRANSACTIONS ON VEHICULAR TECHNOLOGY | IEEE T VEH TECHNOL | 0018-9545 |
| IEEE TRANSACTIONS ON WIRELESS COMMUNICATIONS | IEEE T WIREL COMMUN | 1536-1276 |
| IMAGE AND VISION COMPUTING | IAMGE VISION COMPUT | 0261-8856 |
| INDOOR AND BUILT ENVIRONMENT | INDOOR BUILT ENVIRON | 1420-326X |
| INFORMATION SCIENCES | INFORM SCIENCES | 0020-0255 |
| INFORMATION SYSTEMS | INFORM SYST | 0306-4379 |
| INTERNATIONAL JOURNAL OF ANDROLOGY | INT J ANDROL | 0105-6263 |
| INTERNATIONAL JOURNAL OF CANCER | INT J CANCER | 0020-7136 |
| INTERNATIONAL JOURNAL OF HEAT AND MASS TRANSFER | INT J HEAT MASS TRAN | 0017-9310 |
| INTERNATIONAL JOURNAL OF PRODUCTION ECONOMICS | INT J PROD ECON | 0925-5273 |
| INTERNATIONAL JOURNAL OF REFRIGERATION | INT J REFRIG | 0140-7007 |
| INTERNATIONAL JOURNAL OF THERMAL SCIENCES | INT J THERM SCI | 1290-0729 |
| INVESTIGATIVE RADIOLOGY | INVEST RADIOL | 0020-9996 |
| ISPRS JOURNAL OF PHOTOGRAMMETRY AND REMOTE SENSING | ISPRS J PHOTOGRAMM | 0924-2716 |
| JARO-JOURNAL OF THE ASSOCIATION FOR RESEARCH IN OTOLARYNGOLOGY | JARO-J ASSOC RES OTO | 1525-3961 |
| JOURNAL CONTAMINANT HYDROLOGY | J CONTAM HYDROL | 0169-7722 |
| JOURNAL Of   PHYSICS D-APPLIED PHYSICS | J PHYS D APPL PHYS | 0022-3727 |
| Journal of Advances in Modeling Earth Systems | J ADV MODEL EARTH SY | 1942-2466 |

| JOURNAL OF AGRICULTURAL AND FOOD CHEMISTRY | J AGR FOOD CHEM | 0021-8561 |
|---|---|---|
| JOURNAL OF ALLOY COMPOUNDS | J ALLOY COMPD | 0925-8388 |
| JOURNAL OF ANIMAL ECOLOGY | J ANIM ECOL | 0021-8790 |
| JOURNAL OF CATALYSIS | J CATAL | 0021-9517 |
| JOURNAL OF CHEMICAL THEORY AND COMPUTATION | J CHEM THEORY COMPUT | 1549-9618 |
| JOURNAL OF CHROMATOGRAPHY A | J CHROMATOGR A | 0021-9673 |
| JOURNAL OF CLEANER PRODUCTION | J CLEAN PROD | 0959-6526 |
| JOURNAL OF COMPUTATIONAL PHYSICS | J COMPUT PHYS | 0021-9991 |
| JOURNAL OF DENTISTRY | J DENT | 0300-5712 |
| JOURNAL OF DIFFERENTIAL EQUATIONS | J DIFFER EQUATIONS | 0022-0396 |
| JOURNAL OF ENVIRONMENTAL MANAGEMENT | J ENVIRON MANAGE | 0301-4797 |
| JOURNAL OF FUNCTIONAL ANALYSIS | J FUNCT ANAL | 0022-1236 |
| JOURNAL OF GEOLOGY | J GEOL | 0022-1376 |
| JOURNAL OF GEOPHYSICAL RESEARCH-BIOGEOSCIENCES | J GEOPHYS RES | 0148-0227 |
| JOURNAL OF GEOPHYSICS RESEARCH | J GEOPHYS RES | 0148-0227 |
| JOURNAL OF GLACIOLOGY | J GLACIOL | 0022-1430 |
| JOURNAL OF HEPATOLOGY | J HEPATOL | 0168-8278 |
| JOURNAL OF HYDROMETEOROLOGY | J HYDROMETEOROL | 1525-755X |
| JOURNAL OF IMMUNOLOGY | J IMMUNOL | 0022-1767 |
| JOURNAL OF INFECTIOUS DISEASES | J INFECT DIS | 0022-1899 |
| JOURNAL OF MEMBRANE SCIENCE | J MEMBRANE SCI | 0376-7388 |
| JOURNAL OF METAMORPH GEOLOGY | J METAMORPH GEOL | 0263-4929 |
| JOURNAL OF NEUROLOGY | J NEUROL | 0340-5354 |
| JOURNAL OF NON-CRYSTALLINE SOLIDS | J NON-CRYST SOLIDS | 0022-3093 |
| JOURNAL OF NUCLEAR MEDICINE | J NUCL MED | 0161-5505 |
| JOURNAL OF NURSING MANAGEMENT | J NURS MANAGE | 0966-0429 |
| JOURNAL OF NUTRITION | J NUTR | 0022-3166 |
| JOURNAL OF ORTHOPAEDIC RESEARCH | J ORTHOP RES | 0736-0266 |
| Journal of Physical Oceanography | J PHYS OCEANOGR | 0022-3670 |
| JOURNAL OF PHYSICS G-NUCLEAR AND PARTICLE PHYSICS | J PHYS G NUCL PARTIC | 0954-3899 |
| JOURNAL OF POLYMER SCIENCE PART A-POLYMER CHEMISTRY | J POLYM SCI POL CHEM | 0887-624X |
| JOURNAL OF PROTEOME RESEARCH | J PROTEOME RES | 1535-3893 |
| JOURNAL OF SOL-GEL SCIENCE AND TECHNOLOGY | J SOL-GEL SCI TECH | 0928-0707 |
| JOURNAL OF SOUND AND VIBRATION | J SOUND VIB | 0022-460X |
| JOURNAL OF THE AMERICAN ANCDEMY OF CHILD AND ADOLESCENT PSYCHIATRY | J AM ACAD CHILD PSY | 0890-8567 |
| JOURNAL OF THE AMERICAN CERAMIC SOCIETY | J AM CERAM SOC | 0002-7820 |
| Journal of the American Medical Informatics Association | J AM MED INFORM ASSN | 1067-5027 |
| Journal of the American Society for Information Science and Technology | J AM SOC INF SCI TEC | 1532-2882 |
| JOURNAL OF THE ELECTROCHEMICAL SOCIETY | J ELECTROCHEM SOC | 0013-4651 |
| JOURNAL OF THE OPTICAL SOCIETY OF AMERICA B | J OPT SOC AM B | 0740-3224 |
| JOURNAL OF THE OPTICAL SOCIETY OF AMERICA B-OPTICAL PHYSICS | J OPT SOC AM B | 0740-3224 |
| JOURNAL OF TOXICOLOGY AND ENVIRONMENTAL HEALTH-PART B-CRITICAL REVIEWS | J TOXICOL ENVIRON HEALTH PT B | 1093-7404 |
| JOURNAL OF UROLOGY | J UROLOGY | 0022-5347 |
| KNOWLEDGE-BASED SYSTEMS | KNOWL-BASED SYST | 0950-7051 |

| LAB ON A CHIP | LAB CHIP | 1473-0197 |
|---|---|---|
| LANDSCAPE ECOLOGY | LANDSCAPE ECOL | 0921-2973 |
| LANDSLIDES | LANDSLIDES | 1612-510X |
| LANGMUIR | LANGMUIR | 0743-7463 |
| LASER PHYSICS | LASER PHYS | 1054-660X |
| LASER PHYSICS LETTERS | LASER PHYS LETT | 1612-2011 |
| LIMNOLOGY AND OCEANOGRAPHY | LIMNOL OCEANOGR | 0024-3590 |
| LITHOS | LITHOS | 0024-4937 |
| MACROMOLECULAR BIOSCIENCE | MACROMOL BIOSCI | 1616-5187 |
| MACROMOLECULAR RAPID COMMUNICATIONS | MACROMOL RAPID COMM | 1022-1336 |
| MATERIALS SCIENCE AND ENGINEERING A-STRUCTURES | MAT SCI ENG A-STRUCT | 0921-5093 |
| MICROFLUIDICS AND NANOFLUIDICS | MICROFLUID NANOFLUID | 1613-4982 |
| MICROPOROUS AND MESOPOROUS MATERIALS | MICROPOR MESOPOR MAT | 1387-1811 |
| MOLECUALR AND CELLULAR BIOLOGY | MOL CELL BIOL | 0270-7306 |
| MOLECULAR & CELLULAR PROTEOMICS | MOL CELL PROTEOMICS | 1535-9476 |
| MOLECULAR BIOLOGY AND EVOLUTION | MOL BIOL EVOL | 0737-4038 |
| MOLECULAR CANCER | MOL CANCER | 1476-4598 |
| MOLECULAR ENDOCRINOLOGY | MOL ENDOCRINOL | 0888-8809 |
| MOLECULAR NEUROBIOLOGY | MOL NEUROBIOL | 0893-7648 |
| MOLECULAR PHARMACEUTICS | MOL PHARMACEUT | 1543-8384 |
| MOLECULAR PHARMACOLOGY | MOL PHARMACOL | 0026-895X |
| MOLECULAR PLANT | MOL PLANT | 1674-2052 |
| MOLECULAR THERAPY | MOL THER | 1525-0016 |
| NANO RES | NANO RES | 1998-0124 |
| NANOMEDICINE-NANOTECHNOLOGY BIOLOGY AND MEDICINE | NANOMED-NANOTECHNOL | 1549-9634 |
| NANOSCALE | NANOSCALE | 2040-3364 |
| NANOSCALE RESEARCH LETTERS | NANOSCALE RES LETT | 1931-7573 |
| NANOTECHNOLOGY | NANOTECHNOLOGY | 0957-4484 |
| NATURAL PRODUCT REPORTS | NAT PROD REP | 0265-0568 |
| NEONATOLOGY | NEONATOLOGY | 1661-7800 |
| NEPHROLOGY DIALYSIS TRANSPLANTATION | NEPHROL DIAL TRANSPL | 0931-0509 |
| NEUROBIOLOGY OF DISEASE | NEUROBIOL DIS | 0969-9961 |
| NEW PHYTOLOGIST | NEW PHYTOL | 0028-646X |
| NONLINEAR ANALYSIS-THEORY METHODS & APPLICATIONS | NONLINEAR ANAL-THEOR | 0362-546X |
| NONLINEARITY | NONLINEARITY | 0951-7715 |
| NUCLEIC ACIDS RESEARCH | NUCLEIC ACIDS RES | 0305-1048 |
| NUMERICAL LINEAR ALGEBRA WITH APPLICATIONS | NUMER LINEAR ALGEBR | 1070-5325 |
| NUMERICAL METHODS FOR PARTIAL DIFFERENTIAL EQUATIONS | NUMER METH PART D E | 0749-159X |
| NURSING RESEARCH | NURS RES | 0029-6562 |
| OBSTETRICS AND GYNECOLOGY | OBSTET GYNECOL | 0029-7844 |
| OIL & GAS SCIENCE AND TECNOLOGY | OIL GAS SCI TECHNOL | 1294-4475 |
| ONCOGENE | ONCOGENE | 0950-9232 |
| ONCOTARGET | ONCOTARGET | 1949-2553 |
| OPTIC EXPRESS | OPT EXPRESS | 1094-4087 |

| OPTIC LETTER | OPT LETT | 0146-9592 |
|---|---|---|
| OPTICAL MATERIALS EXPRESS | OPT MATER EXPRESS | 2159-3930 |
| ORGANOMETALLICS | ORGANOMETALLICS | 0276-7333 |
| OSTEOARTHR AND CARTILAGE | OSTEOARTHR CARTILAGE | 1063-4584 |
| PALEOCEANOGRAPHY | PALEOCEANOGRAPHY | 0883-8305 |
| PATTERN RECOGNITION | PATTERN RECOGN | 0031-3203 |
| PHYSICAL CHEMISTRY CHEMICAL PHYSICS | PHYS CHEM CHEM PHYS | 1463-9076 |
| PHYSICS LETTER   B | PHYS LETT B | 0370-2693 |
| PLANT AND SOIL | PLANT AND SOIL | 0032-079X |
| PLOS GENETICS | PLOS GENET | 1553-7404 |
| PLOS ONE | PLOS ONE | 1932-6203 |
| PLOS PATHOGENS | PLOS PATHOG | 1553-7374 |
| POLYM CHEM-UK | POLYM CHEM-UK | 1759-9954 |
| POLYMER | POLYMER | 0032-3861 |
| PRECAMBRIAN RESEARCH | PRECAMBRIAN RES | 0301-9268 |
| PROGRESS IN PHYSICAL GEOGRAPHY | PROG PHYS GEOG | 0309-1333 |
| QUARTERLY JOURNAL OF THE ROYAL METEOROLOGICAL SOCIETY | Q J ROY METEOR SOC | 0035-9009 |
| SCIENCE OF THE TOTAL ENVIRONMENT | SCI TOTAL ENVIRON | 0048-9697 |
| SCRIPT MATERILIA | SCRIPTA MATER | 1359-6462 |
| SENSORS AND ACTUATORS B: CHEMICAL | SENSOR ACTUAT B-CHEM | 0925-4005 |
| SIAM JOURNAL ON APPLIED MATHEMATICS | SIAM J APPL MATH | 0036-1399 |
| SIAM JOURNAL ON COMPUTING | SIAM J COMPUT | 0097-5397 |
| SIAM JOURNAL ON CONTROL AND OPTIMIZATION | SIAM J CONTROL OPTIM | 0363-0129 |
| SIAM Journal on Imaging Sciences | SIAM J IMAGING SCI | 1936-4954 |
| SIAM JOURNAL ON MATHEMATICAL ANALYSIS | SIAM J MATH ANAL | 0036-1410 |
| SIAM JOURNAL ON MATRIX ANALYSIS AND APPLICATIONS | SIAM J MATRIX ANAL A | 0895-4798 |
| SMALL | SMALL | 1613-6810 |
| SMART STRUCTURES AND SYSTEMS | SMART STRUCT SYST | 1738-1584 |
| SOLAR ENERGY | SOL ENERGY | 0038-092X |
| STEM CELLS | STEM CELLS | 1066-5099 |
| SUPERCONDUCTOR SCIENCE & TECHNOLOGY | SUPERCOND SCI TECH | 0953-2048 |
| SURGERY | SURGERY | 0039-6060 |
| TALANTA | TALANTA | 0039-9140 |
| THE CRYOSPHERE | CRYOSPHERE | 1994-0416 |
| THE JOURNAL OF PHYSICAL CHEMISTRY C | J PHYS CHEM C | 1932-7447 |
| TISSUE ENGINEERING | TISSUE ENG | 1076-3279 |
| TRANSPLANTATION | TRANSPLANTATION | 0041-1337 |
| TRANSPORTATION RESEARCH PART E-LOGISTICS AND TRANSPORTATION REVIEW | TRANSPORT RES E-LOG | 1366-5545 |
| ULTRASONICS | ULTRASONICS | 0041-624X |
| ULTRASOUND IN MEDICINE AND BIOLOGY | ULTRASOUND MED BIOL | 0301-5629 |
| WATER RESOURE MANAGEMENT | WATER RESOUR MANAG | 0920-4741 |