# OpenReview forum: "EarthSE: A Benchmark Evaluating Earth Scientific Exploration Capability for Large Language Models"
_ICLR.cc/2026/Conference — ICLR 2026 Poster_

### Official Review · Reviewer_XoSt · 2025-10-18

**Soundness:** 2
**Presentation:** 3
**Contribution:** 2
**Rating:** 4
**Confidence:** 4

**Summary:**

This paper proposes a new set of benchmarks that deal with Earth Science, a topic that is missed by current LLM benchmarks. The contribution is two QA datasets, an easier Earth-Bronze and a more challenging Earth-Silver, with an additional scientific exploration Earth-Gold dataset. All in all, over 100,000 papers were considered for the benchmark construction. The paper additionally proposes a new metric to be used to judge LLM performance on scientific exploration, which is used on Earth-Gold. Lastly, 11 LLMs are evaluated on the 3 proposed benchmarks.

**Strengths:**

- Extending LLMs to topics unexplored by current benchmarks, like Earth Science is important
- The benchmarks are comprehensive, taking into account a pool of more than 100,000 papers. The end result features a large set of 114 disciplines, which is relatively large compared to other similar benchmarks
- The paper is clear and easy to understand

**Weaknesses:**

- LLMs are used in both benchmark construction and in evaluation (retention, win rate). Given that LLMs have been shown to be biased in various ways [1], using them to deconstruct papers into parts and then come up with questions, and then judge proposed answers seems prone to noise, biases and errors. Even with expert curation, the sheer volume of the base set of papers raises concerns with the regards about the amount of errors that can be caught.
- The nature of the tasks means they're hard to define well. For example, the example tasks given in Table 2's research section are open ended and I could see human experts giving varied and even contradictory answers.
- Looking at Earth-Iron/Silver: these feature more well defined answers, but on the other hand, are close to being saturated. Apart from fill in the blank, the best models get 60/70/80% on the other categories. This is approaching saturation, which raises questions about the benchmark’s continued relevance and value to the community.

Minor things:
- Figure 2 should probably be a table
- In line 238 in the caption of Table 2, $P_{hj}$ appears twice
- I think that [2] is very relevant and should be included in the discussion or in Table 1, or both.

[1] Ye, Jiayi, et al. "Justice or prejudice? quantifying biases in llm-as-a-judge." arXiv preprint arXiv:2410.02736 (2024).

[2] Skarlinski, Michael D., et al. "Language agents achieve superhuman synthesis of scientific knowledge." arXiv preprint arXiv:2409.13740 (2024).

**Questions:**

1. How are biases and noise from using LLMs for both question generation and evaluation and metrics controlled? Is it possible that even that the data cleaning step, which uses LLMs, also exhibits biased/erroneous behavior?
2. In open ended tasks, what happens when more than one answer is reasonable? Are there quantitative metrics that show this not to be an issue?
3. Given the near-saturated results on Earth-Bronze/Silver, why are these benchmarks still worth further exploration?

---

> ### Author Response · Authors · 2025-11-19
> **Response to Reviewer XoSt (1/4)**
>
> Dear Reviewer,
>
> Thank you for your positive evaluation and insightful questions. We are pleased that you recognize the comprehensiveness of our benchmark. We have carefully considered your feedback and made revisions accordingly. Below are detailed responses to your questions.
>
> ---
>
> > **Q1: Does using LLMs in data construction and evaluation introduce bias or errors?**
>
> **Response**:
>
> 1. **Reconstruction instead of free creation**: LLMs extract and reconstruct content from original papers, not generating content from scratch. This ensures scientific validity. Earth-Gold questions are fully human-verified. Earth-Iron and Earth-Silver use model difficulty calibration + human inspection. Questions all models answer incorrectly are checked and removed if flawed, ensuring only genuinely challenging questions remain.
>
> 2. **Fine-grained structured construction**: We parse PDFs with MinerU into Markdown, extract section headings, and assign content to abstract, introduction, related work, method, experiment, discussion (optional), and conclusion. Nonconforming papers are manually structured. LLM refinement produces $(M^{0}, L^{0}, M^{1}, L^{1})$ as defined in `Section 3.4`, where $(M^{0}, L^{0})$ comes from related work and $(M^{1}, L^{1})$ from method, experiment, and discussion. Abstract, introduction, and conclusion provide context.
>
> 3. **Contrast-based evaluation ensures objectivity**: Win rate and retention rate are computed by comparing model answers to reference answers, not by direct LLM scoring, which reduces bias. Ranking provides a relatively objective assessment.

---

> ### Author Response · Authors · 2025-11-19
> **Response to Reviewer XoSt (2/4)**
>
> > **Q2: How are open-ended questions evaluated quantitatively?**
>
> **Response**: For the 11 tasks defined in `Table 2` (Earth-Iron and Earth-Silver), we constrained openness and ensured objective evaluation from three aspects: **traceability**, **multi-step reasoning**, and **evaluation metrics**.
>
> 1. **Grounded answers**: In constructing Earth-Iron and Earth-Silver, questions are **directly derived from the source paper**. As described in `Section 3.3` and `Figure 9`, each question is anchored in the content of the paper. This ensures that answers are traceable. For example:
>
> ```
> Question:
> The leading EOF of Eurasian snow depth in late-winter exhibits a ________________ structure, with a band of positive anomalies covering northern Europe and Siberia, and negative anomalies over central Europe, the Himalayas, and north China.
>
> Answer:
> zonally-oriented dipole
> ```
>
> This is a concrete scientific question; the answer is not fully open-ended.
>
> 2. **Verifiable answers**: For complex "research" tasks, our reference answers include full CoT reasoning. As described in `Section 3.3`, the reference includes both the final answer and the derivation steps. `Figure 16` illustrates this. Example:
>
> ```
> Question:
> In the study of ammonia exchange between agricultural cropland and the atmosphere using passive flux samplers, the vertical net flux of ammonia is calculated by dividing the horizontal flux by the weighted mean wind speed at different heights. This calculation helps to establish a concentration gradient. However, the method also accounts for relative variations in concentrations during different wind cycles. How does this method ensure the reduction of errors associated with such variations in wind speeds and ammonia concentrations? By measuring ____, which inherently gives more weight to concentrations during high flux cycles, the method accounts for errors due to time-averaging more effectively than separate measurements of concentration and wind velocity.
>
> Steps:
> Step 1: The method involves measuring horizontal fluxes of ammonia at varying heights, which integrates both concentration and wind speed to determine the vertical net flux.
> Step 2: By focusing on horizontal fluxes, more weight is automatically given to periods with high ammonia concentrations coinciding with high wind speeds, which are periods of high flux.
> Step 3: This integrated measurement approach reduces errors because it avoids separately averaging concentration and wind speed, which can lead to overestimation of flux when using traditional methods.
>
> Answer:
> horizontal fluxes
> ```
>
> 3. **Multi-dimensional evaluation**: For multiple-choice, true/false, and FIB tasks, metrics are explicit and objective. For open-ended questions, we compare answers with reference answers. Semantic similarity measures how close the model’s answer is to the reference. Win rate and similarity together evaluate answer quality. Some model answers differ from reference answers but are high-quality (high win rate), while others are close in wording (high similarity). Example question:
>
> ```
> Question:
> What factors were identified to influence the variability in nodulation across different native sites, particularly for Acacia tortilis, and how do these factors relate to the symbiotic attributes of the provenances tested within the study's experimental framework?
> ```
>
> ```
> Reference Answer:
> The variability in nodulation of Acacia tortilis across different native sites was attributed to temporal and spatial variations, as well as differences in rhizobial population densities. The method of soil collection, whether composite soil from under leguminous canopies or from entire sites, influenced the rhizobial counts. Environmental stressors like high temperatures and low rainfall at specific sites, such as Lodwar riverine, affected rhizobial survival, causing fluctuations in population densities. For A. tortilis, symbiotic attributes varied significantly among its provenances, where the Turkana provenance was most erratic, showing a lack of nodulation at its native site despite a high MPN estimate on one occasion. Such inconsistencies imply that rhizobial populations are influenced by biotic/abiotic factors affecting the provenances over time, making the Mwatate provenance a favorable candidate for future biological nitrogen fixation screenings.
> ```

---

> ### Author Response · Authors · 2025-11-19
> **Response to Reviewer XoSt (3/4)**
>
> ```
> Grok-3's Answer:
> There are significant differences in the quantity and distribution of rhizobia in soils from different regions, mainly influenced by seasonal changes and geographic location. When sampling, soil collected from beneath the canopy of leguminous plants generally contains different rhizobial populations compared to soil taken directly from open areas. In harsher environments, such as hot and arid river valley zones, rhizobial survival is limited, causing fluctuations in their population size. Plant populations from different regions also exhibit notable variation in their symbiotic interactions with rhizobia; in some places, despite abundant rhizobial populations in the soil, nodulation is inconsistent. This indicates that both the vitality of the rhizobia and the adaptability of the plants jointly affect nodulation outcomes. Therefore, the complex interaction between environmental factors and microbial ecology determines the regional differences in nodulation capacity. Plants from certain locations demonstrate more stable symbiotic performance and are thus recommended as primary candidates for future biological nitrogen fixation research and applications.
> ```
>
> ```
> Gemini-2.5-Flash's Answer:
> Nodulation differences in Acacia tortilis across various native habitats are influenced not only by the quantity and distribution of rhizobia but also strongly by the plant’s genetic adaptability and root system morphology. Populations from different origins show significant variation in root characteristics such as taproot length, lateral root number, and root hair density. These structural traits determine how effectively the plant can interact with rhizobia, thereby impacting both the number and quality of nodules formed. Some populations with well-developed lateral roots are able to establish symbiotic relationships with more rhizobia, promoting enhanced nodulation.
>
> Furthermore, the physicochemical properties of soil play a critical role in regulating rhizobial activity and their symbiotic relationship with plants. Differences in soil pH, salinity, organic matter content, and nutrient status across sites collectively affect rhizobial survival and reproduction. Soil environments that are highly saline or deficient in organic matter typically restrict rhizobial populations and their activity, consequently influencing nodulation performance. Thus, even when rhizobial numbers are relatively high in certain soils, unfavorable environmental conditions or incompatibility between the plant population and rhizobia can limit nodulation capacity.
>
> The relationship between rhizobial abundance and nodulation is not simply a direct positive correlation. Some sites have high rhizobial densities, yet the nodulation rate remains low due to ineffective signal recognition or rejection responses between the plants and rhizobia. Conversely, plant populations with strong genetic adaptability can achieve better nodulation efficiency through highly effective symbiotic mechanisms, even in soils with lower rhizobial densities. This underscores the decisive role of the plant’s genetic background in successful rhizobial symbiosis.
>
> In summary, the variation in nodulation ability among Acacia tortilis populations results from the interaction between plant genetic traits and soil environmental factors, including rhizobial population structure and physicochemical conditions. Populations exhibiting strong genetic adaptability and superior symbiotic capacity, such as those from the Mwatate region, are more suitable candidates for future biological nitrogen fixation screening. This also highlights the necessity of considering both plant genetics and soil microbial ecology in tandem to achieve optimal nitrogen fixation outcomes in research and practical applications.
> ```
>
> Grok-3’s answer has higher semantic similarity (0.84) but weaker than the reference. Gemini-2.5-Flash is judged better than the reference but has lower similarity (0.69). These two modes show that **win rate and similarity jointly evaluate answer quality**, allowing objective assessment for open-ended questions regardless of novel or similar viewpoints.
>
> In summary, the openness of our dataset is strictly constrained and does not lead to divergent or subjective evaluation results.

---

> ### Author Response · Authors · 2025-11-19
> **Response to Reviewer XoSt (4/4)**
>
> > **Q3: For Earth-Iron / Earth-Silver: These subsets have more clear-cut answers, but they seem close to saturation. Why are these benchmarks still valuable?**
>
> **Response**:
>
> 1. **Comprehensive analysis**: The large volume of questions in Earth-Iron (from 100k papers) covers a wide range of Earth science subfields and multiple tasks (e.g., scientific computation, code generation, fill-in-the-blank). This broad coverage allows us to diagnose LLM weaknesses across many fine-grained tasks.
>
> 2. **Skill imbalance**: As shown in our paper (`Figure 1`), although some models perform well in code generation tasks, the **scientific computation** task average accuracy remains below 60%. Furthermore, for **FIB (fill-in-the-blank)** tasks, Earth-Iron average score is 14.50%, and Earth-Silver average score is 11.06%. These numbers indicate that current models still have clear limitations in certain foundational scientific tasks.
>
> 3. **Future research directions**: These evaluation results provide clear guidance for model improvement (e.g., enhancing numerical reasoning, fill-in-the-blank abilities). Earth-Iron and Earth-Silver serve as diagnostic benchmarks, helping advance LLM capabilities in Earth sciences and fundamental scientific reasoning.
>
> Therefore, despite apparent saturation in some tasks, these two subsets remain highly valuable resources.
>
> ---
>
> > **Q4: Minor issues – references / writing**
>
> **Response**: Thank you for pointing these out. We will provide a complete table of disciplines in the supplementary material. Other issues have been corrected in the revised PDF. We also added discussion on LLM data construction and evaluation in the appendix with corresponding citations.
>
> ---
>
> We sincerely hope these responses clarify your concerns. If our revisions and explanations address your questions, we kindly request you to consider updating your score. Please feel free to reach out with further questions. Thank you again for your careful and constructive review.

---

> > ### Comment · Reviewer_XoSt · 2025-11-19
> >
> > Thanks for your answers, I've raised my score accordingly. Please include the clarifications and examples in the final manuscript.

---

> > > ### Author Response · Authors · 2025-11-20
> > >
> > > Thank you very much for updating your score. We have revised the PDF on OpenReview according to your suggestions and plan to include more complete cases in the appendix to improve the final version of the paper. We sincerely appreciate your constructive feedback once again.

---

### Official Review · Reviewer_zDSD · 2025-10-27

**Soundness:** 3
**Presentation:** 3
**Contribution:** 3
**Rating:** 4
**Confidence:** 3

**Summary:**

This paper presents EarthSE, a QA dataset benchmarking the earth science exploration capability of LLMs. These questions are sourced from 100k Earth Science papers, and are constructed into three subsets (Iron, Silver, and Gold). The authors decomposed each paper and then used GPT-4 to generate dialogues, followed by using human expert validation to ensure the dialogue quality. On several LLMs, the authors found that CoT guidance enhances the performances on challenging questions.

**Strengths:**

- This is the first science QA dataset for earth science that is derived from this magnitude of academic papers.
- This paper presents a principled approach to construct benchmark datasets from earth science publications.

**Weaknesses:**

- Using one metric (SES) to assess advanced capabilities of scientific exploration (e.g., methodology induction, limitation analysis, and concept proposal) seems a bit too on reductionist side to me. Looking at the two components of the SES metric (retention rate and diversity), I became less convinced about the utility of this metric to the intended capabilities to evaluate.
- How are the Iron and the Silver subsets divided? The boundary seems a bit arbitrary to me.
- Similarly, the decision to make Iron & Silver in QA formats while making Gold in dialogue format also appears very arbitrary to me.

**Questions:**

- How is the paper decomposed into the structured components? Is it also using GPT-4?
- The experts scored the dialogue, but did they score a subset of the QA questions in the benchmark?

---

> ### Author Response · Authors · 2025-11-19
> **Response to Reviewer zDSD (1/2)**
>
> Dear Reviewer,
>
> Thank you for your positive evaluation and insightful questions. We are pleased that you recognize the value of using large-scale data in our work. We have carefully considered your feedback and made revisions accordingly. Below are detailed responses to your questions.
>
> ---
>
> > **Q1: Is the SES (Scientific Exploration Score) metric too simplified?**
>
> **Response**: Thank you for your comment. When designing SES, we considered its limitations and made the following clarifications and enhancements:
>
> 1. **Rationale of SES**: The SES metric is carefully designed, selecting two dimensions: **retention (r)** and **diversity (d)**, which correspond to two core aspects of scientific exploration: quality and novelty.
>
>    - **Retention (r)**: Measures how well model-generated answers are ranked relative to highly-cited reference answers, reflecting quality, accuracy, and depth.
>    - **Diversity (d)**: Measures the divergence of answers obtained via multiple samplings in semantic space, reflecting exploration and innovation breadth.
>
> 2. **Avoiding fully subjective evaluation**: Retention (r) is derived from comparative ranking among responses, reflecting **relative quality** rather than an absolute score of a single answer, thus avoiding direct subjective LLM scoring. Diversity is computed numerically, **avoiding subjectivity**. Together, these two dimensions provide an objective measure of model capability in open-ended scientific exploration.
>
> 3. **Introducing LLM Judge metrics**: We additionally include direct LLM scoring to assess responses along more dimensions. Using **Gemini 2.5 Pro**, we scored responses on two new dimensions:
>
>    - **Logical coherence** (0~10): Whether the limitations identified in Turn 1 are addressed in the new proposals of Turn 2.
>    - **Feasibility** (0~10): Whether proposed methods include detailed, reproducible steps or experimental protocols.
>
>    | Model              | Logical Coherence | Feasibility |
>    | ----------------- | --------------------- | ---------------- |
>    | GPT‑4o            | 6.41                  | 5.11             |
>    | Claude-3.7-Sonnet | 6.93                  | 5.77             |
>
> 4. **Terminology revision**: In the revised PDF, we replaced "advanced capabilities" with the more precise phrase **"depth & diversity in scientific exploration."**
>
> ---
>
> > **Q2: Earth-Iron / Earth-Silver subset division**
>
> **Response**:
>
> 1. **Data source differences**: The division is based on clear, objective differences in the source corpora. As detailed in `Section 3.2`, Earth-Iron is drawn from a broad $P_{base}$ corpus (100,000 papers), designed for comprehensive evaluation. In contrast, Earth-Silver is drawn from $P_{hj}$ (10,000 papers), a pre-filtered corpus containing only high-impact journals (see `Table 7` for examples). This division is principled: **comprehensive coverage (Iron)** vs. **specialized knowledge (Silver)**.
>
> 2. **Difficulty validation**: This principled division is supported by experimental results (`Table 4`). As shown in `Section 4.2`, LLM performance drops sharply from Earth-Iron to Earth-Silver. For example, the average win rate in free-response Q&A drops from 72.86% (Iron) to 41.15% (Silver). This significant increase in difficulty indicates that our high-impact journal selection ($P_{hj}$) successfully isolates more challenging, specialized knowledge. Earth-Silver contains fewer questions, so for evaluating performance across multiple tasks and domains, larger sets like Earth-Iron are more suitable.
>
> 3. **Similar practices in prior work**: Similar dataset divisions are common, e.g., ClimaQA `[1]` (ICLR 2025) uses ClimaQA-Gold and ClimaQA-Silver, supporting the reasonableness of such a design.
>
> 4. **Evaluation objectives**: In summary, Earth-Iron and Earth-Silver differ in evaluation objectives (comprehensive vs. specialized), construction (100,000 broad papers vs. 10,000 high-impact papers), and evaluation outcomes (overall metrics vs. challenging scores), demonstrating the intentionality and necessity of our dataset design.

---

> ### Author Response · Authors · 2025-11-19
> **Response to Reviewer zDSD (2/2)**
>
> > **Q3: Division of QA vs dialogue format**
>
> **Response**:
>
> 1. **From basic understanding to reflection and innovation**: The different evaluation formats reflect the varying capabilities being assessed, from foundational knowledge understanding to higher-order reasoning and creativity. For basic skills, LLM domain knowledge and ability to explain scientific phenomena can be **effectively assessed with standard Q&A**. Moreover, QA is an efficient method for evaluating foundational capabilities.
>
> 2. **Simulating real scientific scenarios**: Earth-Gold’s dialogue format formalizes scientific exploration tasks and is a core innovation of our benchmark. As defined in `Section 3.1`, we model scientific exploration as an iterative process: $(M^{i+1},L^{i+1})=LLM(M^{i},L^{i})$, naturally requiring multi-turn interaction. As shown in `Figure 6`, Turn 1 evaluates analysis of $(M^{0}, L^{0})$ (existing work), and Turn 2 evaluates proposals for $(M^{1}, L^{1})$ (new work). This **iterative, critique-based process** cannot be assessed through single-turn Q&A.
>
> ---
>
> > **Q4: How is paper decomposition (structuring) implemented?**
>
> **Response**: This is an excellent question. When structuring papers, we consider their sectional organization and use rule-based parsing combined with LLM refinement for efficient and accurate structuring:
>
> 1. **Rule-based parsing**: MinerU parses the PDF into Markdown. Using Markdown syntax, first- and second-level headings are extracted, and content is assigned to sections: abstract, introduction, related work, method, experiment, discussion (optional), conclusion. Papers that do not follow these rules are manually structured by human experts.
>
> 2. **LLM refinement**: The LLM refines each section to produce the four parts defined in `Section 3`.4, $(M^{0}, L^{0}, M^{1}, L^{1})$, where $(M^{0}, L^{0})$ (existing work) mainly comes from related work, and $(M^{1}, L^{1})$ (new work) mainly comes from method, experiment, and discussion sections. Abstract, introduction, and conclusion provide context, ensuring the model understands the overall research direction while focusing on specific sections.
>
> This two-stage decomposition ensures that the four parts reflect genuine structural differences in the papers. The LLM performs only textual summarization without complex structural synthesis or free generation, maximizing rigor and fidelity. Implementation details are included in the supplementary material.
>
> ---
>
> > **Q5: Did you perform human review or spot-checks on questions in Iron/Silver subsets?**
>
> **Response**: Yes. Earth-Iron and Earth-Silver use difficulty calibration + human inspection for quality control:
>
> 1. **Difficulty calibration**: LLMs attempt the questions to remove those most models answer correctly and to identify those all models answer incorrectly. Human experts review the latter, removing questions with errors and retaining genuinely challenging questions. Remaining questions have at least one model answering correctly, ensuring reliable construction.
>
> 2. **Spot-checking**: Additionally, humans checked 5% of Earth-Iron and 10% of Earth-Silver questions and found no issues, confirming dataset reliability.
>
> ---
>
> We sincerely hope these responses clarify your concerns. If our revisions and explanations address your questions, we kindly request you to consider updating your score. Please feel free to reach out with any further questions. Thank you again for your careful and constructive review.
>
> ---
>
> > References
>
> 1. *ClimaQA: An Automated Evaluation Framework for Climate Question Answering Models*

---

> > ### Comment · Reviewer_zDSD · 2025-11-19
> >
> > Thanks for the response. These clarified much of my confusion. I have raised my score to 6.

---

> > > ### Author Response · Authors · 2025-11-20
> > >
> > > Thank you for updating your score. We truly appreciate your thoughtful feedback and encouraging comments!

---

### Official Review · Reviewer_aJ8Z · 2025-11-02

**Soundness:** 3
**Presentation:** 3
**Contribution:** 3
**Rating:** 8
**Confidence:** 3

**Summary:**

EarthSE introduces a benchmark for evaluating LLMs’ Earth science exploration capabilities. It includes three datasets: (1) Earth-Iron, (2) Earth-Silver, and (3) Earth-Gold. These datasets target different levels of knowledge, from broad assessment (Earth-Iron) to higher difficulty requiring professional understanding (Earth-Silver), and finally open-ended scientific exploration through dialogue (Earth-Gold). The authors then analyze these capabilities in leading LLMs and find that they perform reasonably well on Earth-Iron but struggle with Earth-Silver and show low retention and diversity on Earth-Gold tasks.

**Strengths:**

EarthSE introduces a benchmark with various difficulty levels and broad coverage across subfields (114 disciplines and 11 tasks), as well as multi-turn, open-ended dialogues for scientific exploration and discovery. To build the QA datasets, the authors leverage around 100,000 research papers and categorize them by journal impact, citation count, and topical focus. They prioritize high-quality sources and exploit paper structure aligned with the scientific discovery process, followed by extensive data cleaning and expert validation. They also introduce additional metrics for assessing Earth-Gold, evaluate multiple LLMs, and report interesting, important insights.

**Weaknesses:**

(1) Earth-Gold uses a fixed two-turn format. Real scientific exploration often needs longer iterative chains. It would be interesting to see the results with more turns.

(2) The inference-time “initial CoT steps” taken from question construction seem to boost FIB accuracy, but may leak answer-related cues a model wouldn’t get at test time. It would be interesting to see the self-generated CoT results as well.

**Questions:**

(1) In lines 455-460, it is mentioned that the initial CoT steps from the question construction step are provided during inference to see whether that helps with performance on more challenging question types. However, it is not clear to me why you didn’t instead ask the model to generate CoT reasoning while answering the question. Providing the CoT steps from the question construction step might leak knowledge about the answer that it shouldn’t, and that the model wouldn’t normally have access to.

---

> ### Author Response · Authors · 2025-11-19
> **Response to Reviewer aJ8Z**
>
> Dear Reviewer,
>
> Thank you for your positive evaluation and insightful questions. We are pleased that you recognize that our evaluation provides interesting and important insights. We have carefully considered your feedback and made revisions accordingly. Below are detailed responses to your questions.
>
> ---
>
> > **Q1: Earth-Gold uses two-turn dialogues, whereas real scientific exploration often involves longer iterative processes.**
>
> **Response**: Your point is very valuable, and we would like to provide additional clarification:
>
> 1. **High-efficiency evaluation**: The current two-turn design (first turn for summarization + second turn for proposal) was chosen considering construction costs (reviewers, time, manpower) and controllability, yet it still effectively simulates the typical **"research → analysis → innovation"** workflow in scientific work. In other words, the two-turn dialogue represents the most efficient and reasonable evaluation approach we could implement.
>
> 2. **Future work**: We plan to extend Earth-Gold to 3–4 turns or more to better reflect the iterative nature of scientific exploration. Additionally, we will study model performance in multi-turn interactions with respect to consistency, creativity, and stability.
>
> ---
>
> > **Q2: The initial chain-of-thought (CoT) reasoning provided during question construction may leak the answer. Why not let the model generate its own CoT?**
>
> **Response**: Thank you for this critical comment. We conducted additional experiments and analysis:
>
> 1. **Self-generated CoT experiment**: We allowed models (without using our provided CoT prompts) to generate their own chain-of-thought and answer the questions. Results show limited performance improvement (e.g., DeepSeek-V3 FIB accuracy increased from **12.80%** to **13.48%**, +0.68%), indicating that self-generated CoT alone cannot fully compensate for deficiencies in complex scientific reasoning.
>
>    | Model             | Without CoT | With CoT |
>    | ----------------- | ----------- | -------- |
>    | DeepSeek-V3       | 12.80       | 13.48    |
>    | GPT-4o            | 18.40       | 19.82    |
>
> 2. **Guidance vs Leakage**: The CoT provided during question construction does **not directly expose the answer**; it serves as a reasoning scaffold `[1]`. Experiments show that different models benefit differently from the same scaffold, reflecting their **instructability**. While the scaffold provides guidance, it does not reveal answer-specific information. Moreover, when using the same CoT prompt, models show varying performance improvements: those with larger gains require only minimal guidance to answer correctly, indicating better scaffold utilization, which is an interesting observation.
>
> ---
>
> We sincerely hope these responses clarify your concerns. If our revisions and explanations address your questions, we kindly ask you to consider updating your score. Please feel free to reach out with any further questions. Thank you again for your careful and constructive review.
>
> ---
>
> > References
>
> 1. *Reasoning Scaffolding: Distilling the Flow of Thought from LLMs*

---

### Official Review · Reviewer_h6Tz · 2025-11-03

**Soundness:** 3
**Presentation:** 4
**Contribution:** 3
**Rating:** 6
**Confidence:** 4

**Summary:**

The paper introduces a suite of 3 new LLM benchmarks, two QA benchmarks around Earth science.
There are two QA style datasets and a multi-turn conversation benchmark.

The benchmark is based on 100k papers. Papers are categorized for field/sub-field.
The authors select subsets of size 100k, 10k, 1k by impact/popularity.


The QA pairs are constructed using LLMs. Each question/answer pair is built around a task category. The tasks include understanding but also reasoning and research tasks, such as experimental design and code generation.
For each paper tasks are selected to match the abstract.
Too simple examples are filtered.
Experts reviewed the remaining questions, keeping


For the multi-turn dialog they model papers as a transition from( existing methods and their limitations) to (novel method and its constraints). They extract these from the top 1k most cited papers.
Then they auto-generate two turn dialogues of summary / proposal with the respective limitations.
Again, the examples are reviewed by experts.

To evaluate the multi-turn task, the paper proposes a novel metric based on ranking several samples from an LLM against the gold example along with a diversity metric.

A slightly dated set of models is used in experiments showing significant differences between models, particularly for the multi-step task.

There are some issues with this paper that I am hopeful can be fixed.

**Strengths:**

Addressed a variety of scientific tasks, particularly beyond reasoning/extraction

Large corpus of papers used.

Paper is well written and has very little fluff.

Headroom for evaluation

Expert verification of at least parts of the data

**Weaknesses:**

GPT-4o was used in several steps of the generation process which might lead to bias. It doesn't look like a problem from the results but this should at least be explicitly acknowledged.

There is little detail on "human experts determined their retention based on the question's value". Can you tell a bit more about this process?

Evaluation of the Free Response QA tasks could be extended beyond embedding similarity, e.g. by using a judge model or using rubrics.

It is unclear if diversity should be evaluated by temperature sampling. Without calibration, that parameter might do different things with different models, as evidenced by results in Table 5

Evaluation of some more recent models with thinking enabled

**Questions:**

Is there an expert review of only the questions or also of the answers? How were the experts recruited / compensated? What is the acceptance / rejection rate? What is the agreement for these decisions?


Smaller issues / improvements

Please check references for the models, e.g.
* Not sure Islam & Moushi is the right reference for GPT-4o
* Gemini is cited as "Team et al" ?

The model names need to be more explicit, e.g. Gemini 2.0 Flash or Claude 3.7 Sonnet

In Figure 3, "a task that best suits your construction…" did you want to say qualification?

---

> ### Author Response · Authors · 2025-11-19
> **Response to Reviewer h6Tz (1/2)**
>
> Dear Reviewer,
>
> Thank you very much for your careful review and constructive comments. We are pleased that you recognize the quality of our paper writing. We have carefully considered your feedback and made revisions accordingly. Below are detailed responses to your questions.
>
> ---
>
> > **Q1: Using GPT-4o multiple times in the construction process may introduce bias.**
>
> **Response**: We fully understand this concern. To reduce potential bias, we employed the following mechanisms and provided detailed explanations:
>
> 1. **Strict data filtering and multi-stage quality control**: The construction process does not solely rely on GPT-4o, but also includes rule-based filtering, semantic validation, difficulty calibration, and **human expert review**. This multi-stage process minimizes inaccuracies or biases potentially introduced by LLM generation.
>
> 2. **Reconstruction from papers rather than free-form creation**: GPT-4o is used to **extract** content from the original papers and then reconstruct Q&A pairs or dialogues, rather than freely generating content unrelated to the papers. This design ensures high consistency with real scientific materials.
>
> 3. **Consistency with prior studies**: Using LLMs to synthesize training or evaluation data has been widely adopted recently. Surveys indicate that LLM-driven synthetic data can effectively alleviate the scarcity of annotated resources `[1]`, and recent systematic reviews specifically discuss LLM-driven data generation, curation, and evaluation pipelines `[2]`.
>
> 4. **Benefits for benchmark fairness and reproducibility**: Using a single high-capability model (e.g., GPT-4o) to construct all instances ensures consistent question style and difficulty. This uniformity is crucial for evaluating other models, as it ensures all models are tested on the same “scale.”
>
> 5. **Add a new appendix section**: We have added an analysis of potential biases and limitations of using LLM for data construction to the `Appendix A.5` of the revised PDF. Thank you again for your suggestions.
>
> ---
>
> > **Q2: How were expert evaluations conducted to determine question retention based on value?**
>
> **Response**: Thank you for pointing this out. In the revised PDF, we have added detailed expert scoring information:
>
> 1. We invited **Master’s/PhD experts in Earth sciences** (e.g., meteorology, geology) to evaluate each candidate question across **six dimensions**:
>
>    - Scientific Method: Does it involve concrete scientific methods (e.g., data processing, analysis, experiments)?
>    - Data Source: Does it include specific datasets or experimental data?
>    - Temporal/Spatial Context: Does it reference a specific event, time, or location (e.g., earthquake, typhoon)?
>    - Literature Citation: Does it cite a specific scientific paper?
>    - Method Details: If proposing a new method, is it specific (step-by-step, named, detailed)?
>    - Limitation Analysis: Does it provide logical, in-depth discussion of method limitations?
>
> 2. Each dimension was scored 0/1/2 (0 = absent, 1 = present but insufficiently detailed, 2 = very specific). Only questions with **at least 3 dimensions > 0** and a **total score ≥ 5** were retained. This ensures that retained questions are scientifically valuable, specific, and challenging.
>
> 3. We added a table in the revised PDF to clearly illustrate this process:
>
> | Dimension           | Description                              | Score Range |
> | ------------------ | ---------------------------------------- | --------- |
> | Scientific Method   | Does it involve data analysis/experiments? | 0–2       |
> | Data Source         | Are datasets or sources explicitly specified? | 0–2     |
> | Temporal/Spatial Context | Does it reference a specific scientific event? | 0–2 |
> | Literature Citation | Is a scientific paper cited?             | 0–2       |
> | Method Details      | Are methods described step-by-step or named? | 0–2    |
> | Limitation Analysis | Is there in-depth discussion of limitations? | 0–2    |

---

> ### Author Response · Authors · 2025-11-19
> **Response to Reviewer h6Tz (2/2)**
>
> > **Q3: Suggestion to include LLM judges or rubric for free-response tasks.**
>
> **Response**: We agree with your suggestion. Initially, we did not use LLMs for scoring due to two limitations: first, scores could be subjective without an anchor; using the reference answer as an anchor is more objective. Second, LLM scoring tends to be conservative, limiting score differentiation. Nevertheless, combining LLM judgments with semantic similarity can provide a new valuable evaluation dimension. We introduce the following enhancement:
>
> 1. **LLM-as-Judge**: Using Gemini-2.5-Pro to score model responses along two dimensions:
>
>    - Specificity: 0 (very vague) – 10 (very specific)
>    - Expertise: 0 (common knowledge) – 10 (highly professional)
>
>    On the Earth-Silver subset, we obtained:
>
> | Model               | Specificity | Expertise |
> | ------------------ | ----------- | -------- |
> | GPT‑4o             | 7.05        | 6.92     |
> | Claude-3.7-Sonnet  | 6.79        | 6.41     |
>
> 2. **Composite metric**: LLM scores are combined with embedding similarity (weighted / multi-dimensional) to provide a more comprehensive quality assessment.
>
> ---
>
> > **Q4: Using fixed temperature (temperature = 0.6)**
>
> **Response**: Thank you for this critical question. Below is a more detailed rationale and analysis:
>
> 1. **Choice of temperature 0.6**: We refer to moderate temperatures commonly used in generation tasks `[3,4,5]`. Specifically, prior work `[3]` demonstrates that 0.6 often provides superior performance across multiple tasks. Thus, using 0.6 in our study is reasonable both experimentally and academically.
>
> 2. **Sensitivity analysis**: We compared sampling temperatures of 0.4, 0.6, and 0.8. Temperature 0.6 achieves a good balance between creativity and output quality. Sampling temperature affects diversity and output quality, influencing retention rate metrics. The table below shows retention, diversity, and SES for Gemini-2.5-Flash at different temperatures:
>
> | Sampling temperature | Retention | Diversity | SES      |
> | ------------------ | --------- | -------- | -------- |
> | 0.4                | 55.71     | 1.8794   | 1.1780   |
> | 0.6                | 50.56     | 2.7016   | **1.3710** |
> | 0.8                | 31.43     | 3.4215   | 1.0871   |
>
> 3. **Metric rationale**: This also highlights the rationality of SES design, which constrains evaluation via both diversity and retention, preventing simple model hacks.
>
> 4. **Future improvements**: We plan to explore advanced sampling strategies (e.g., selective or risk-aware sampling) to balance quality and diversity dynamically.
>
> ---
>
> > **Q5: Suggest evaluating thinking-enabled models.**
>
> **Response**: Thank you for the suggestion. We added tests on the following newer models:
>
> | Model             | Earth-Silver FIB |
> | ---------------- | ---------------- |
> | DeepSeek-R1      | 17.82%           |
> | o4-mini          | 19.41%           |
> | Qwen-3-8B        | 11.79%           |
>
> Results indicate that even these newer models face significant challenges on specialized Earth science tasks, further highlighting the benchmark's discriminative capability.
>
> ---
>
> > **Q6: Expert review implementation details**
>
> **Response**: Thank you for raising these important details. We provide the following clarifications:
>
> 1. **Review scope**: Experts evaluated both questions and answers.
> 2. **Recruitment & compensation**: We recruited 12 experts with Earth science backgrounds (Master’s/PhD) via a professional annotation company. Each review took ~5 minutes, compensated at \\$1 per 5 minutes (\\$12/hour).
> 3. **Pass rate & consistency**: The pass rate was ~44%. To ensure scoring consistency, we held tri-party meetings (annotators, company, team) to define scoring rules. Each item was scored by at least two experts; Cohen’s κ ≈ 0.72.
>
> ---
>
> > **Q7: Minor issues – model references / naming / figure phrasing**
>
> **Response**: Thank you for these detailed suggestions. We corrected the references, clarified model names, and revised phrasing in figures in the revised PDF.
>
> ---
>
> We sincerely hope these responses clarify your concerns. If our explanations and revisions address your questions, we kindly request you to consider updating your score. Please feel free to reach out with further questions. Thank you again for your careful and constructive review.
>
> ---
>
> > References
>
> 1. *On LLMs-Driven Synthetic Data Generation, Curation, and Evaluation: A Survey*
> 2. *Synthetic Data Generation Using Large Language Models: Advances in Text and Code*
> 3. *Piloting Copilot, Codex, and StarCoder2: Hot Temperature, Cold Prompts, or Black Magic?*
> 4. *Fast on the Easy, Deep on the Hard: Efficient Reasoning via Powered Length Penalty*
> 5. *Reconsidering Overthinking: Penalizing Internal and External Redundancy in CoT Reasoning*

---

### Author Response · Authors · 2025-11-30
**Rebuttal Summary**

**Dear Area Chair,**

As stated in the official ICLR email, *“The ICLR organizers have been notified of an OpenReview bug that leaked the names of authors, reviewers, and area chairs for all conferences.”* In light of this, the decision by the ICLR organizers to *“reverting scores”* in order to maintain fairness in the review process is entirely reasonable. We sincerely appreciate the efforts of the ICLR organizers to uphold the integrity and fairness of the review process. At the same time, **we are deeply grateful to the Area Chairs and all others involved who will be undertaking additional work as part of this process**.

However, as authors who always uphold academic ethics, **we regret that the scores reflecting our rebuttal (all of which were positive) are not directly visible**. In this regard, we have summarized our rebuttal and the process through which reviewers adjusted their scores below. We respectfully request that you take these positive evaluations from the reviewers into consideration.

# Rebuttal Summary

## Overview

Our work introduces the first large-scale, rigorously structured benchmark for scientific reasoning in Earth science, covering 114 disciplines and 11 task types, addressing a critical gap in current LLM evaluation. **Our rebuttal successfully resolved major reviewer concerns.** Two initially borderline reviewers (zDSD and XoSt) raised their scores from **4 → 6** after reading our response. The other two (h6Tz and aJ8Z) remain positive, making the overall panel stance more favorable toward acceptance.

| Reviewer | Initial | After Rebuttal | Notes|
| - | - | - | - |
|**h6Tz**|6|**6**| Accept|
|**aJ8Z**|8|**8**| Strongly supportive |
|**zDSD**|4|**6**| Concerns addressed  |
|**XoSt**|4|**6**| Concerns addressed  |

---

## Reviewer h6Tz

**Initially rated 6**, with concerns about GPT-4o's role, expert question retention, and free-response evaluation. Our rebuttal clarified:

1. Multi-stage pipeline (rule-based filtering, semantic checks, difficulty calibration, expert scoring) ensures GPT-4o reconstructs, not freely generates, content.
2. Expert review protocol: six scoring dimensions, ~44% pass rate, dual annotators, κ ≈ 0.72.
3. Added LLM-as-judge evaluation (Gemini-2.5-Pro) integrated with semantic metrics.
4. Temperature sensitivity analyses (0.4/0.6/0.8) justify SES and diversity evaluation.
5. Experiments with thinking-enabled models (DeepSeek-R1, o4-mini, Qwen3-8B).

The reviewer did not post any new discussions or update his/her rating before the system locked rating. **The final rating remained a positive 6.**

---

## Reviewer aJ8Z

**Initially rated 8**, praising benchmark breadth and evaluation. Concerns:

1. Two-turn design may miss iterative scientific reasoning.
2. Potential answer leakage from CoT reasoning in question construction.

Rebuttal clarified:

1. Two-turn setup balances evaluation efficiency and reflects the “research → analysis → innovation” workflow; multi-turn extensions are planned.
2. Experiments with self-generated CoT showed minor gains, confirming construction CoT acts as scaffold, not answer leak.

The reviewer did not post any new discussions or update his/her rating before the system locked rating. **The final rating remains a strong support of 8.**

---

## Reviewer zDSD

Initially rated 4, questioning SES metric, Earth-Iron vs. Earth-Silver split, and QA/dialogue formats. Rebuttal clarified:

1. SES combines retention and diversity; supplemented with LLM judge metrics.
2. Earth-Iron vs. Earth-Silver split is corpus-driven; difficulty confirmed by performance drop.
3. QA/dialogue formats reflect different evaluation goals.
4. Two-stage paper decomposition plus human review ensures quality.

Score increased **4 → 6; stance now positive**.

---

## Reviewer XoSt

Initially rated 4, concerned about LLM biases, open-ended tasks, and apparent saturation. Rebuttal clarified:

1. LLMs reconstruct, do not freely generate; human verification ensures quality.
2. Structured pipeline (PDF parsing, section extraction, LLM refinement) maintains objectivity.
3. Multi-dimensional evaluation ensures robust scoring for open-ended tasks.
4. Benchmarks remain diagnostic despite high Earth-Iron/Silver performance.
5. Minor issues addressed in references, tables, and appendix.

Score increased **4 → 6; stance now positive**.

---

**All of our responses are detail-oriented and supported by experimental data or case studies. Moreover, we promptly incorporated these new experimental results and additional detailed explanations into the PDF on OpenReview.** This demonstrates our diligence and effort during the rebuttal phase, as well as our careful consideration of all reviewers’ comments

Finally, we would like to once again express our heartfelt thanks to all reviewers for their careful evaluations, and to the Area Chairs and related staff for their tremendous contributions in maintaining the fairness of the review process.

Sincerely,

The Authors

---

### Meta-Review · Area_Chair_6SKf · 2025-12-27

**Summary:**

The paper presents EarthSE, a dedicated benchmark for assessing LLMs in Earth science exploration, encompassing three hierarchically designed datasets: Earth-Iron, Earth-Silver, and Earth-Gold. Tailored to distinct knowledge depth requirements, Earth-Iron focuses on broad-based competency evaluation, Earth-Silver targets advanced tasks demanding professional domain understanding, and Earth-Gold emphasizes open-ended scientific inquiry via interactive dialogue.

 All key considerations—including the design of two-turn dialogue, the formulation of evaluation metrics, and the recruitment and compensation of human experts—have been appropriately addressed. I therefore recommend accepting this paper, with the suggestion that greater emphasis be placed on the human expert component within the ethics statement.

**Reviewer Concerns:**

All reviewer concerns are adequatly addressed.

**Reviewer Scores:**

Reviewer XoSt and Reviewer zDSD may raise the score.

---

### Decision · Program_Chairs · 2026-01-26

Accept (Poster)